

# Enhancing Winter Climate Simulations of the Great Lakes: Insights from a New Coupled Lake-Ice-Atmosphere (CLIAv1) Model on the Importance of Integrating 3D Hydrodynamics with a Regional Climate Model

Pengfei Xue[1,2,3], Chenfu Huang[2], Yafang Zhong[4], Michael Notaro[4], Miraj B. Kayastha[1], Xing Zhou[1], Chuyan Zhao[2], Christa Peters-Lidard[5], Carlos Cruz[5], Eric Kemp[5]

[1] Department of Civil, Environmental and Geospatial Engineering, Michigan Technological University, Houghton, MI, USA

[2] Great Lakes Research Center, Michigan Technological University, Houghton, MI, USA

[3] Environmental Science Division, Argonne National Laboratory, Lemont, IL, USA

[4] Nelson Institute Center for Climatic Research, University of Wisconsin-Madison, Madison, WI, USA

[5] National Aeronautics and Space Administration Goddard Space Flight Center, Greenbelt, MD, USA

*Correspondence to*: Pengfei Xue, pexue@mtu.edu





## Abstract

The Laurentian Great Lakes significantly influence the climate of the Midwest and Northeast
United States, due to their vast thermal inertia, moisture source potential, and unique heat and
moisture flux dynamics. This study presents a newly developed coupled lake-ice-atmosphere
(CLIAv1) modeling system for the Great Lakes by coupling the National Aeronautics and
Space Administration (NASA)-Unified Weather Research and Forecasting (NU-WRF) regional
climate model (RCM) with the three-dimensional (3D) Finite Volume Community Ocean
Model (FVCOM) and investigates the impact of coupled dynamics on simulating the Great
Lakes' winter climate. By integrating 3D lake hydrodynamics, CLIAv1 addresses the
limitations of traditional one-dimensional (1D) lake and demonstrates superior performance in
reproducing observed LSTs, ice cover distribution, and the vertical thermal structure of the
Great Lakes compared to the NU-WRF model coupled with the default 1D Lake Ice Snow and
Sediment Simulator (LISSS). CLIAv1 also enhances simulation of over-lake atmospheric
conditions, including air temperature, wind speed, and sensible and latent heat fluxes,
underscoring the importance of resolving complex lake dynamics for reliable climate
projections. More importantly, this study addresses the crucial question about what are the key
processes influencing lake thermal structure and ice cover that are missed by 1D lake models
but effectively captured by 3D lake models. Through process-oriented numerical experiments,
we identify key 3D hydrodynamic processes—ice transport, heat advection, and shear
production in turbulence—that explain the superiority of 3D lake models over 1D lake models,
particularly in cold season performance and lake-atmosphere interactions. Properly resolving
these processes using 3D hydrodynamic model is crucial for successfully simulating the lake-
ice-atmosphere coupled Great Lakes winter system. This research underscores the necessity of
incorporating 3D hydrodynamic models in RCMs to improve our predictive understanding of
the Great Lakes' response to climate change. The findings advocate for a shift towards high-
resolution, physics-based modeling approaches to ensure accurate future climate and
limnological projections for large freshwater systems.



## 1 Introduction

The Laurentian Great Lakes, with a surface area of 246,000 km², represent Earth's largest
surface freshwater resources, containing 21% of the world's surface freshwater and 84% of
North America's surface freshwater (Botts and Krushelnicki, 1987; EPA, 2014; Notaro et al.,
2015; Xue et al., 2022). Over 55 million people live within the Great Lakes' megaregion
(Todorovich, 2009; Sharma et al., 2018). The lakes support the United States' and Canadian
economies by impacting drinking water supply, shipping, fishing, power production,
transportation, manufacturing, wastewater treatment, agriculture, and recreation (Vaccaro and
Read, 2011). The Great Lakes' support of these vital industries sustains approximately 1.3
million jobs and $82 billion in annual wages (Rau et al., 2020). As an invaluable resource to
wildlife and society, the ecologically diverse Great Lakes Basin is home to over 3,500 animal
and plant species, including over 170 fish species (Botts and Krushelnicki, 1987; Crossman and
Cudmore, 1998; EPA, 2014). The basin's wetlands serve as spawning and nesting habitat,
reduce erosion, and protect water quality (Notaro et al., 2015).
The Great Lakes are critically important in terms of their impacts on the climate of the Midwest
and Northeast United States and southern Ontario, Canada. The regional climate is highly
sensitive to the Great Lakes due to the lakes' vast thermal inertia, potential source of moisture
to the overlying atmosphere, and contrasts in heat, moisture, roughness, and albedo compared
to surrounding land (Changnon and Jones, 1972; Scott and Huff, 1996; Chuang and Sousounis,
2003; Notaro et al., 2013a; Briley et al., 2021; Wang et al., 2022). During late autumn through
winter, when cold, dry continental air masses from Canada pass over the relatively mild Great
Lakes, the air masses are destabilized and moistened, leading to enhanced cloud cover and
precipitation downwind of the lakes (Niziol et al., 1995; Ballentine et al., 1998; Kristovich and
Laird, 1998; Notaro et al., 2013b; Shi and Xue, 2019). During the broader unstable lake season,
which spans from September to March and is characterized by amplified lake-effect cloud
cover and precipitation due to lake surface temperatures typically exceeding overlying air
temperatures, lake-effect snowfall typically peaks during December-January, and lake ice cover
is most extensive during February-March (Assel, 1990; Niziol et al., 1995; Kristovich and
Laird, 1998; Lam and Schertzer, 1999; Notaro et al., 2013b). The establishment of extensive
lake ice cover usually by mid-late winter dampens over-lake turbulent fluxes of heat and





moisture, subsequently reducing resulting lake-effect precipitation (Brown and Duguay, 2010;
Notaro et al., 2021). Specifically, increasing lake ice cover leads to a linear reduction in latent
heat fluxes and nonlinear reduction in sensible heat fluxes (Gerbush et al., 2008). When
relatively cool (warm) air masses pass over the Great Lakes during winter (summer), the
relatively warm (cool) lake surface reduces (enhances) atmospheric stability and increases
(decreases) deep convection, cloud cover, and precipitation (Scott and Huff, 1996; Holman et
al., 2012; Bennington et al., 2014). The lakes' relatively low roughness compared to the
surrounding land leads to strengthened over-lake wind speeds and potential shoreline
convergence in support of enhanced lake-effect precipitation. Due to the lakes' large thermal
inertia and resulting seasonal evolution in lake-air temperature contrast, the Great Lakes
typically strengthen wintertime cyclones and summer anticyclones and weaken summertime
cyclones and wintertime anticyclones (Notaro et al., 2013a). The basin is a preferred zone of
wintertime cyclogenesis due to the relative warmth of the lake surfaces and consequential
enhancement in low-level convergence (Petterssen and Calabrese, 1959; Colucci, 1976;
Eichenlaub, 1978).
Given the aforementioned substantial influence of the Great Lakes on regional climate, their
representation and evaluation in both global and regional climate models have been the focus of
several studies in the past decade. There is a wide spectrum among climate models regarding
the treatment of large lakes. Due to their coarse spatial resolution, most global climate models
(GCMs), including those from various phases of the Coupled Model Intercomparison Project
(CMIP), either omit the Great Lakes entirely or offer a crude representation using wet soil,
wetlands, ocean grid cells, or 1D lake models (Briley et al., 2021; Minallah and Steiner, 2021).
Among regional climate models (RCMs) without lake models, many apply a rudimentary
approach to estimate lake surface temperatures (LSTs) by extrapolating the closest ocean grid
cell's sea-surface temperatures (SSTs), likely from Hudson Bay or the North Atlantic Ocean,
from the initial and lateral boundary conditions datasets to the lake grid cell, potentially
inducing vast biases and intra-lake discontinuities in LST and ice cover (Gao et al., 2012;
Mallard et al., 2015; Spero et al., 2016; Xiao et al., 2016; Hanrahan et al., 2021). This approach
is the default treatment of LSTs in the Weather Research and Forecasting (WRF) model
(Hanrahan et al., 2021; Wang et al., 2022). Alternatively, the WRF Preprocessing System can



designate time-averaged 2-m air temperatures to the underlying lake surfaces to provide
estimated lower boundary conditions of LST based on the user-specified time window for
temporal averaging and time lag for addressing thermal inertia (Wang et al., 2012; Mallard et
al., 2015; Hanrahan et al., 2021; Wang et al., 2022). However, this approach still produces
unrealistic LSTs and ice cover as the lakes cannot achieve equilibrium with the overlying
atmosphere due to the lack of interactive lake-atmosphere feedbacks (Bullock et al., 2014;
Spero et al., 2016).
For those GCMs and RCMs that aim to incorporate coupled lake-atmosphere interactions, most
apply 1D lake models (Perroud et al., 2009; Martynov et al., 2010; Stepanenko et al., 2010;
Subin et al., 2012). Those include 2-layer bulk models founded in similarity theory such as the
Freshwater Lake (FLake) model (Mironov et al., 2010), thermal diffusion models which
parameterize eddy diffusivity such as the Minnesota Lake Water Quality Management Model
(MINLAKE, Riley and Stefan, 1988) and the Hostetler model (Hostetler and Bartlein, 1990),
Lagrangian turbulence models such as the Dynamics Reservoir Simulation Model (DYRSM,
Yeates and Imberger, 2003), and $k-\epsilon$ turbulence closure models with horizontally averaged
velocity such as LAKE (Stepanenko and Lykossov, 2005; Stepanenko et al., 2011) and Simstrat
(Goudsmit et al., 2002). Each of these different categories of 1D lake models has its own
advantages and disadvantages (Perroud et al., 2009; Martynov et al., 2010; Stepanenko et al.,
2010; Subin et al., 2012). As demonstrated in these studies, the deficiencies include struggles
with simulating seasonal stratification in FLake, insufficient mixing for deep lakes in the
Hostetler model, and excessive mixing for shallow lakes in the computationally expensive
turbulence models.
Multiple modeling studies have assessed the performance of coupling RCMs to 1D lake models
in the Great Lakes region. While this coupling permits the representation of key lake-
atmosphere interactions and the heterogeneous spatiotemporal patterns of LSTs and lake ice
cover, 1D lake models typically perform poorly at reproducing the lake thermal structure and
seasonal ice evolution of large, deep lakes, such as Lake Superior, due to the overly simplified
hydrodynamic processes. Common biases in 1D lake models include an anomalously early
timing of both spring-summer stratification and autumn turnover, with positive biases in
summer LST and negative biases in winter LST (Bennington et al., 2014; Mallard et al., 2014).

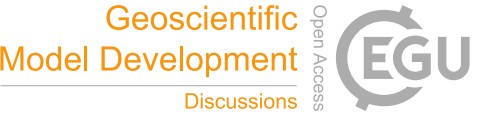

The International Centre for Theoretical Physics (ICTP) Regional Climate Model version 4
(RegCM4), coupled to the 1D Hostetler lake model, yields a prolonged lake ice season with
excessive ice cover due to the neglect of horizontal heat advection within the lakes (Notaro et
al., 2013b). The coupling of a thermal diffusion lake model, the Lake, Ice, Snow and Sediment
Simulator (LISSS, Subin et al., 2012) to the WRF model (available starting with version 3.6 of
WRF) results in an early warm-up and overly rapid cool-down in the seasonal evolution of
LSTs for deep lakes, along with an early onset of lake ice cover in support of its excessive
abundance (Xiao et al., 2016). Mallard et al. (2014) found that WRF, coupled to FLake,
produced the best performance for Lake Erie (the smallest and shallowest Great Lake) and the
worst performance for Lake Superior (the largest and deepest Great Lake) among the Great
Lakes in terms of simulated LST and ice cover biases. Often, modelers aim to reduce biases in
the simulated vertical temperature profile of deep lakes in 1D models by artificially enhancing
the vertical eddy diffusivity to crudely compensate for the absence of a dynamic circulation and
vertical mixing processes (Subin et al., 2012; Bennington et al., 2014; Lofgren, 2014; Gu et al.,
2015; Mallard et al., 2015), although such a non-physics based approach may only yield limited
benefits to minimizing these biases (Xiao et al., 2016). The lack of fully resolved lake
hydrodynamics in models, including dynamic 3D lake circulation, upwelling and downwelling,
thermal bar formation, explicit horizontal mixing, and ice motion, along with overly simplified
stratification processes, unrealistic treatment of eddy diffusivity, and the assumption of
instantaneous mixing of thermal instabilities (Song et al., 2004; Martynov et al., 2010, 2012;
Stepanenko et al., 2010; Bennington et al., 2014; Gu et al., 2015; Mallard et al., 2015; Sharma
et al., 2018; Sun et al., 2020; Notaro et al., 2021; Hutson et al., 2024) has been the main
obstacle in further improving climate simulations for the Great Lakes Basin.
In recent years, a limited number of Great Lakes studies have aimed to enhance the
representation of three-dimensional (3D) lake hydrodynamical processes and reduce the
substantial biases in LST and ice cover associated with 1D lake models by coupling RCMs with
3D hydrodynamic models (Xue et al., 2017, 2022; Sun et al., 2020; Kayastha et al., 2023).
These studies have responded to the urgent call for continued progress in coupling high-
resolution RCMs with 3D lake models that address the complex processes and features of large,
deep lakes, as highlighted in previous research (Martynov et al., 2010; Bennington et al., 2014;
Briley et al., 2021; Notaro et al., 2021). Xue et al. (2017) developed a two-way coupled 3D



lake-ice-climate modeling system, known as the Great Lakes-Atmosphere Regional Model
(GLARM), by coupling RegCM4 with a 3D unstructured-grid hydrodynamic model, the Finite
Volume Community Ocean Model (FVCOM, Chen et al., 2012). The resulting coupled 3D
modeling system exhibited notable skill in reproducing the mean, variability, and trends in
regional climate across the Great Lakes Basin and the physical characteristics of the Great
Lakes, including their thermal structure and ice cover, significantly improving upon previous
RCM experiments coupled with 1D lake models. The updated version, GLARM-V2, has been
utilized to generate future climatic and limnological projections for the Great Lakes region
(Xue et al., 2022). Similarly, Sun et al. (2020) developed a lake-atmosphere-hydrology
modeling system by coupling the Climate-WRF (CWRF) model with 3D FVCOM and
compared its performance against CWRF coupled with the 1D LISSS. They found that the
former configuration outperformed the latter in simulating LST, ice cover, and the vertical
thermal structure in the Great Lakes. Kayastha et al. (2023) developed and validated the WRF-
FVCOM Two-way Coupling (WF2C) model, showing WF2C improved upon past 1D lake
model-based studies by significantly reducing the simulated summer LST bias, and revealing
how coupled lake-atmosphere dynamics can influence summer LST by modifying surface heat
fluxes through impacts on meteorological state variables. These studies underscore the
advantages of coupling an RCM with a 3D lake hydrodynamic model for accurately depicting
lake physical processes and lake-atmosphere feedbacks in the Great Lakes Basin. However,
there is a notable absence of research dedicated to identifying the fundamental processes
resolved in 3D lake models that contribute to these improvements, which is important to
optimize effort allocation in future model development and improve our predictive
understanding of the system. This knowledge gap is particularly significant for the Great Lakes
during the winter seasons.
This paper attempts to address this knowledge gap, by developing a new coupled lake-ice-
atmosphere (CLIA version 1 or CLIAv1) modeling system for the Great Lakes by coupling the
National Aeronautics and Space Administration (NASA)-Unified Weather Research and
Forecasting (NU-WRF) regional climate model (RCM) with the three-dimensional (3D) Finite
Volume Community Ocean Model (FVCOM). Note that CLIAv1 is hereinafter referred to as
NU-WRF/FVCOM for the sake of particular attention given to comparing NU-WRF's
performance during the cold season when two-way coupled with 3D FVCOM (NU-



WRF/FVCOM) versus 1D LISSS (NU-WRF/LISSS). After a thorough validation of the
coupled model, we conduct a series of process-oriented numerical experiments to identify the
most important hydrodynamic processes that contribute to the superiority of the 3D lake model
over the 1D lake model in enhancing lake-atmosphere coupling for the Great Lakes.

## 204    2    Model, Data, and Numerical Experiment Design

### 205    2.1    Atmosphere Model

NU-WRF is an observation-driven integrated regional modeling system, developed at NASA's
Goddard Space Flight Center (GSFC), that resolves chemistry, aerosol, cloud, precipitation and
land processes at satellite-resolvable scales (roughly 1–25 km) to improve the continuity
between microscale, mesoscale and synoptic processes. Developed as a superset of the
community WRF, NU-WRF unifies the NCAR - Advanced Research version of WRF model
(WRF-ARW) with the GSFC Land Information System (LIS, Kumar et al., 2006; Peters-Lidard
et al., 2007, 2015), the Goddard Chemistry Aerosol Radiation and Transport (GOCART) model
(Chin et al., 2000), the Goddard radiation and microphysics schemes (Shi et al., 2014), and the
Goddard Satellite Data Simulator Unit (G-SDU, Matsui et al., 2013, 2014). NU-WRF
simulations here utilize the Noah Land Surface Model, which simulates soil moisture and
temperature, skin temperature, snowpack depth and the energy flux and water flux terms of the
surface energy balance and surface water balance (Mitchell, 2005). Currently, by default, the
two-way lake-atmosphere interactions in NU-WRF are represented using the embedded 1D
LISSS (Subin et al., 2012) from the Community Land Model version 4.5 (Oleson et al., 2013)
with modifications by Gu et al. (2015).
Notaro et al. (2021) conducted 20 simulations to identify the regionally optimal NU-WRF
configuration and schemes for the cold season period of November 2014-March 2015 in the
Great Lakes region. The best model configuration was referred to as the "Morrison
combination" and is used in this study. The "Morrison combination" includes Morrison
microphysics (Morrison et al., 2009), Rapid Radiative Transfer Model (RRTM, Mlawer et al.,
1997) longwave radiation physics, Community Atmosphere Model (CAM, Collins et al., 2004)
shortwave radiation physics, Mellor-Yamada-Nakanishi-Niino Level 2.5 (MYNN2.5,
Nakanishi and Niino, 2006, 2009) planetary boundary layer physics, and Mellor-Yamada-



Nakanishi-Niino (MYNN, Nakanish, 2001) surface layer schemes. The improved simulations
of air temperature and surface insolation using the Morrison combination primarily benefits
from the Community Atmosphere Model's shortwave radiation scheme (Notaro et al., 2021).
The Morrison combination is essentially the WRF configuration determined by Mooney et al.
(2013) to produce the best simulated wintertime temperature simulation over Europe, who
found that winter air temperatures are highly sensitive to the choice of radiation physics.
The NU-WRF one-way nested configuration consists of an outer domain with 15-km grid
spacing for the majority of North America and an inner domain with 3-km grid spacing for the
Great Lakes region (Fig. 1), with the atmospheric vertical resolution assigned to 61 levels. The
initial and lateral boundary conditions are provided by the Global Data Assimilation System 0-
hour analysis. The cumulus parameterization option used for the outer domain is the Kain-
Fritsch scheme (Kain and Fritsch, 1990; Kain, 2004) with resolved, unparameterized
convection in the inner domain.

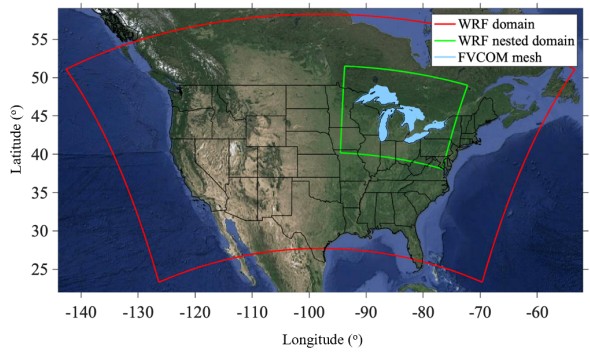

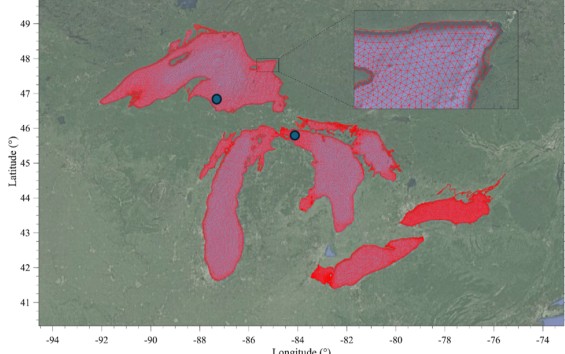


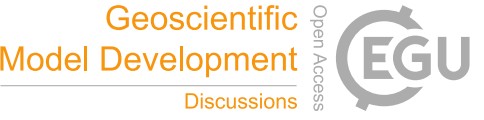

**Figure 1.** NU-WRF nested domains (upper panel) and unstructured mesh used in FVCOM to represent
the Great Lakes in FVCOM (lower panel). The two dots denote the locations of Granite Island (87.4°W,
46.7°N) on Lake Superior and Spectacle Reef (84.1°W, 45.7°N) on Lake Huron.
**2.2    Hydrodynamic Model**
The hydrodynamic model, FVCOM, is a free-surface, primitive equation hydrodynamic model
that solves the momentum, continuity, temperature, salinity, and density equations and is closed
physically and mathematically using turbulence closure submodels (Chen et al., 2012).
Numerically, FVCOM employs the finite-volume method over an unstructured triangular grid
and vertical sigma layers, optimizing flexibility and accuracy for complex terrains. The grid
resolution adjusts from 1–2 km near coasts to resolve coastal geometry complexity, to 2-4 km
offshore to improve computational efficiency (Fig. 1), with the model comprising 35,000 grid
cells and 40 sigma layers. Vertical mixing processes are modeled using the Mellor–Yamada
level-2.5 (MY25) turbulence closure model (Mellor and Yamada, 1982), while horizontal
diffusivity is derived from velocity shear and grid resolution through the Smagorinsky (1963)
formulation.
FVCOM also includes an unstructured-grid, finite-volume version of the Los Alamos
Community Ice Code (CICE), which describes ice thickness distribution in time and space.
CICE includes a thermodynamic model to compute local growth rates of snow and ice due to
vertical conductive, radiative, and turbulent fluxes. It also features an ice dynamics model to
simulate the ice pack velocity due to wind and ice-water stress, Coriolis effects, sea surface
slope, and internal stress, estimated with elastic–viscous–plastic rheology (Hunke and
Dukowicz, 1997). The transport model in CICE calculates the advective process of the areal
concentration, ice volumes, and other state variables. The ridging parameterization in CICE
addresses mechanical redistribution, which transfers ice among thickness categories (Hunke et
al., 2010).
In contrast, the default 1D lake model, LISSS, embedded in NU-WRF solves the 1D thermal
diffusion equation (i.e. lake thermal dynamics only) by segmenting the vertical stratification of
the lake into multiple distinct levels that include: snow (applicable when the snow's thickness
surpasses a specified minimum value); the combined section of lake water and ice, collectively



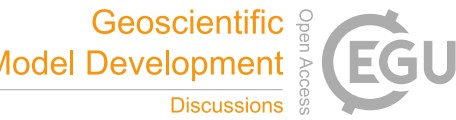

identified as the "lake body"; and the bottom layers consisting of sediment, soil, and bedrock
(collectively termed "sediment" unless specified differently). This structured division allows for
simulating thermal dynamics within each segment, facilitating a prediction of temperature
distribution and variations across the lake's depth (Subin et al., 2012).
**3    Two-way Coupling of NU-WRF/FVCOM**
The development of interactively coupled model systems [see review by Giorgi and Gutowski
Jr. (2015)] emerged quickly in the late 2000s driven by rapid technological advancement and
the increase in computational capability. Model coupling is essential to multi-physics
simulations representing various components of the Earth system. Over the past two decades,
several coupling technologies for earth system modeling have been developed. Examples
include the Earth System Modeling Framework (ESMF), the Model Coupling Toolkit (MCT),
and the OASIS-MCT coupler, which is the latest version of the OASIS3 coupler interfaced with
the Model Coupling Toolkit (MCT) that offers a fully parallel implementation of coupling field
regridding and exchange (Valcke et al., 2012; Craig et al., 2017). Although coupling
implementations can follow different approaches, their applications in geophysical simulations
typically carry out several key functions, including interpolating and transferring the coupling
fields between different model grids, managing data transfer between constitutive models at a
desired coupling frequency, and coordinating the execution of the constituent models in a
parallel computational environment (Valcke et al., 2012). In general, coupling data must be
interpolated and transferred between the constituent models under several constraints, such as
conservation of physical properties, numerical stability, consistency with physical processes,
and computational efficiency.
In the study, NU-WRF and FVCOM are run simultaneously, exchanging information
bidirectionally at 1-hour intervals through the OASIS3-MCT coupler. FVCOM dynamically
calculates the LST and ice cover, providing these as overlake surface boundary conditions to
NU-WRF. Meanwhile, NU-WRF calculates and supplies the atmospheric forcings required by
FVCOM, including surface air temperature, surface air pressure, relative and specific humidity,
total cloud cover, surface winds, and downward shortwave and longwave radiation.



### 3.1 Data for Model Validation

The average daily LST, obtained from composite images taken by the Advanced Very High Resolution Radiometer, is sourced from version 2 of the Great Lakes Surface Environmental Analysis (GLSEA) LST Dataset, developed by the National Oceanic and Atmospheric Administration's (NOAA) Great Lakes Environmental Research Laboratory (GLERL). A comprehensive evaluation carried out by Schwab et al. (1999) shows that LST measurements from GLSEA and the buoy-based LSTs had an average discrepancy of less than 0.5°C across all buoys, with a root-mean-square difference (RMSD) between 1.10°C and 1.76°C. The Great Lakes Ice Cover Dataset, compiled by GLERL, has also been added to the GLSEA product. The dataset incorporates daily average ice cover data across the lakes, which draws from ice products produced by the United States National Ice Center and the Canadian Ice Service, and is detailed in studies by Assel et al. (2002, 2013), Assel (2005), and Wang et al. (2012).

In-situ lake thermistor measurements for vertical lake thermal structure were obtained from Spectacle Reef on Lake Huron (Fig. 1). Measurements for over-lake atmospheric variables, including air temperature, wind velocity, downward shortwave radiation, and sensible and latent heat fluxes, were obtained from Granite Island on Lake Superior and Spectacle Reef on Lake Huron through the Great Lakes Evaporation Network (GLEN) (Blanken et al., 2011; Spence et al., 2011; Lenters et al., 2013; Spence et al., 2013; Spence et al., 2019). These level-1 eddy covariance data received minimal adjustments, notably the elimination of heat spikes and a basic visual quality assessment. This dataset was compared with an independent dataset of Great Lakes' turbulent fluxes developed by Moukomla and Blanken (2017), revealing a "good statistical agreement" between them, with RMSD ranging from 4.5 to 7 W/m$^2$ for latent and sensible heat fluxes (Moukomla and Blanken, 2017).

### 3.2 Design of Numerical Experiments

We designed numerical experiments in two categories. In category 1, we evaluate the cold season performance of the NU-WRF/FVCOM two-way coupling (case C1-1) against the NU-WRF/LISSS 1D lake model (case C1-2). To ensure the objectivity of the comparison, both C1-1 and C1-2 utilize an identical NU-WRF configuration (except for differences in lake treatment) as described in Section 2.1, following the optimal NU-WRF configuration for the study region as determined by Notaro et al. (2021). The comparison of C1-1 and C1-2 aims to



examine the overall impact of using a 3D versus a 1D lake model configuration on simulating
lake hydrodynamic conditions and the subsequent impact on the atmospheric state through
lake-ice-atmosphere interactions from November 2014 to March 2015. The initial lake
conditions of November 2014 were obtained from multiple years of FVCOM standalone
simulations driven by Climate Forecast System Reanalysis (CFSR) forcing Xue et al. (2015).
In category 2, a set of process-oriented numerical experiments is designed to identify the
impact of various 3D hydrodynamical processes critical to the coupled Great Lakes system.
These processes are either neglected or oversimplified by the NU-WRF/LISSS 1D lake model
while being resolved by the NU-WRF/FVCOM 3D lake model. Case C2-1 (NoIceTransp) is
designed to examine the impact of ice transport associated with currents (Section 5.1). In this
scenario, FVCOM is configured identically to C1-1, except that ice dynamics, ice velocity
fields, and ice pack transport are disabled in FVCOM. Instead, only ice thermal dynamics are
simulated to account for the spatio-temporal evolution of ice thickness distribution through
thermodynamic growth and melting processes (Bitz and Lipscomb, 1999). Consequently, the
ice model is simplified to function as an energy-conserving thermodynamic model, akin to that
used in the 1D lake model.
Case C2-2 (NoHeatAdv) analyzes the impact of 3D heat transport associated with lake
circulation. FVCOM is configured identically to C1-1, except that the advective heat transport
associated with current movement is disallowed in C2-2. This is realized by turning off the
advection terms in the temperature equation in FVCOM, which is essentially an advection-
diffusion equation that governs the distribution and evolution of temperature (Section 5.2).
Therefore, the temperature calculation is simplified to imitate the 1D vertical diffusion equation
used in the 1D lake model.
Case C2-3 (NoShearProd) aims to assess the influence of 3D currents on calculation of
turbulent mixing, a crucial factor in controlling the heat redistribution and thermal structure in
the lakes. In this case, we exclude the turbulence shear production term that depends on
currents in the turbulent kinetic equation (Section 5.3). In summary, the three cases in category
2 collectively reveal the significant impacts of currents in elements that are not accounted for in
the LISSS 1D lake model, i.e. on ice transport, heat transport, and turbulent mixing intensity,



359 respectively. These experiments are summarized in Table 1.

360 **Table 1.** A summary of the numerical model experiments. The "3D currents" column shows if the
361 experiment resolves the 3D currents of the Great Lakes. The "Ice transport" column shows if the
362 experiment resolves the ice transport associated with currents in the Great Lakes. The "Heat advective
363 transport" column shows if the experiment resolves the 3D heat transport associated with Great Lakes
364 circulation. The "Shear production in turbulence" column shows if the experiment uses the turbulence
365 shear production term that depends on currents in the turbulent kinetic equation. The "Lake model"
366 column shows the lake model used in the experiment.

| Experiment | 3D currents | Ice transport | Heat advective transport | Shear production in turbulence | Lake model |
|---|---|---|---|---|---|
| C1-1 (Lake3D) | Yes | Yes | Yes | Yes | FVCOM |
| C1-2 (Lake1D) | No | No | No | No | LISSS |
| C2-1 (NoIceTransp) | Yes | No | Yes | Yes | FVCOM |
| C2-2 (NoHeatAdv) | Yes | Yes | No | Yes | FVCOM |
| C2-3 (NoShearProd) | Yes | Yes | Yes | No | FVCOM |

367

368 **4 Results**

369 **4.1 Lake Temperature and Ice Coverage**

370 The NU-WRF/FVCOM model (case C1-1) accurately captures the seasonal evolution of LSTs
371 across all of the lakes with lake-mean LST root-mean-square-error (RMSE) less than 0.4°C
372 (Fig. 2 upper panels). During November, the lakes are in the middle of their cooling period and
373 the LSTs decrease rapidly, yet at different paces, largely due to variations in the lakes' depth



and latitude, which leads to strong spatial heterogeneity in LST (Fig. 3, left panels). The
GLSEA data and the 3D lake model closely align in terms of the spatial LST patterns, with
warmer waters of 10-12°C in the central and eastern basins of Lakes Erie and Ontario and 8-
10°C in the southern basins of Lakes Michigan and Huron, while much cooler temperatures are
found across Lake Superior, ranging between 4-6°C. The most notable underestimation of LST
by the 3D lake simulation occurs in the southern basin of Lake Huron, while the model well
captures the LSTs in the northern basin of Lake Huron. Transitioning to January 2015 (Fig. 3,
right panels), at the onset of the ice season, NU-WRF/FVCOM accurately reflects the seasonal
cooling of the lakes, showing a significant reduction in LSTs, while also well delineating the
detailed temperature differences between the colder nearshore and relatively warmer offshore
waters, in good agreement with the observational data. On the other hand, NU-WRF/LISSS
(case C1-2) fails to capture the spatial heterogeneity in LSTs, but also generates a systematic
cold bias of 2-3°C during January across nearly all of the lakes (Fig. 3, bottom panels). Such a
cold bias was persistent in the NU-WRF/LISSS (Lake1D) simulation throughout the cold
season, as detailed in Notaro et al. (2021).

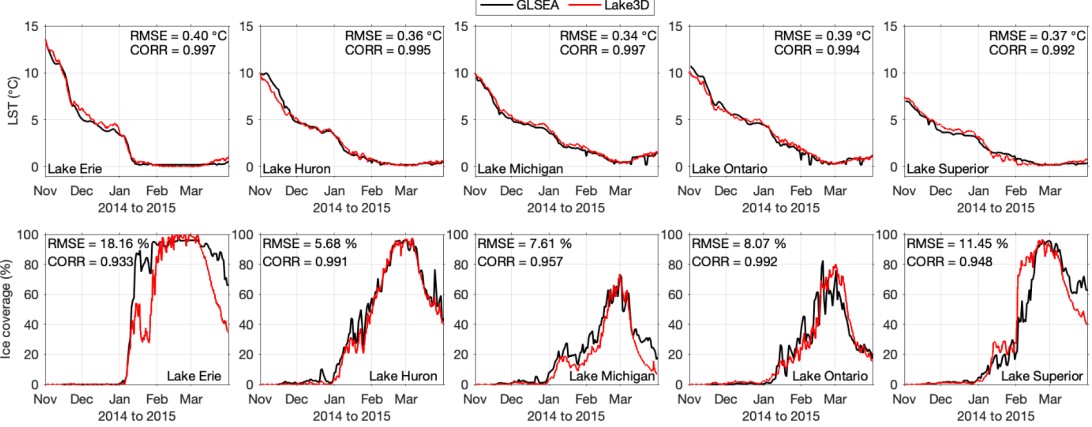


**Figure 2.** Time series of daily lake-averaged LST (°C, upper panels) and percent ice cover (lower panels)
for the five lakes from GLSEA data (black lines) and NU-WRF/FVCOM 3D lake model simulations (red
lines) during the simulation period of November 2014-March 2015. Both the temporal correlation and
RMSE are reported in each panel.

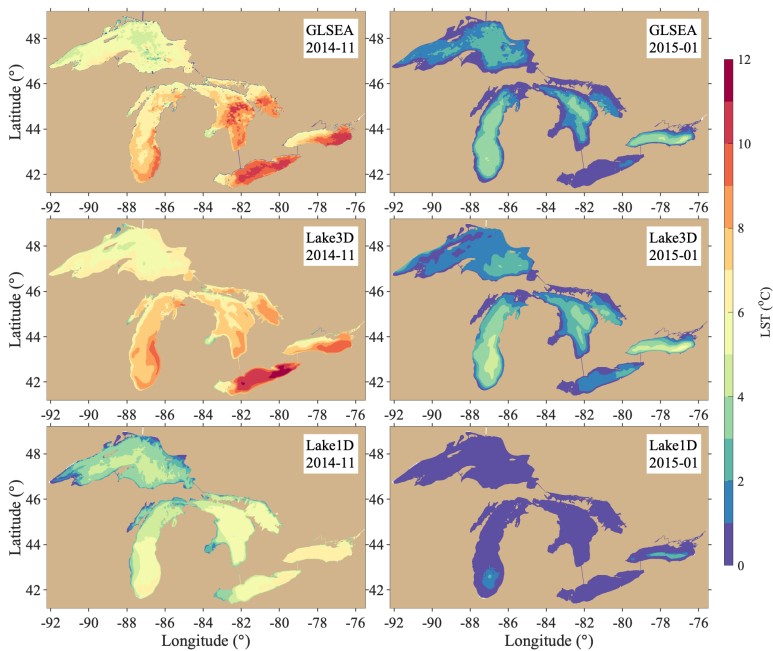

**Figure 3.** Spatial patterns of monthly mean LSTs (°C) from GLSEA data (top panels), NU-WRF/FVCOM 3D lake model simulations (middle panels), and NU-WRF/LISSSS 1D lake model simulations (bottom panels) for November 2014 (left panels) and January 2015 (right panels).

NU-WRF/FVCOM (Lake3D) also demonstrates its skill in capturing the evolution of the vertical thermal structure within the lake, which is particularly challenging in large and deep lakes (Bennington et al., 2014; Xue et al., 2017). As exemplified in Fig. 4, the in-situ thermistor measurement at Spectacle Reef on Lake Huron is located in a deep region with a water depth greater than 200 meters. The 3D model reproduces the conclusion of the summer stratification process until the end of November. The following turnover, a seasonal process where the surface water cools, becomes denser, and sinks—mixing with the warmer water from below— is also represented in the 3D lake model between December and January. Subsequently, the winter inverse stratification, where colder water (< 4ºC) lies above warmer water due to the fact that freshwater's density peaks at 4ºC, is captured by the 3D model as it develops from February onward, although the model shows a stronger winter inverse stratification and earlier onset than observed. In contrast, NU-WRF/LISSS falls short of these detailed observations. Not only does it mispredict the occurrence of turnover and winter stratification much earlier than



observed, but it also substantially underestimates the extent of mixing between the surface and
deeper waters. This underestimation results in a flawed representation of excessive surface
cooling and a substantial overestimation of the warming of the deep waters.

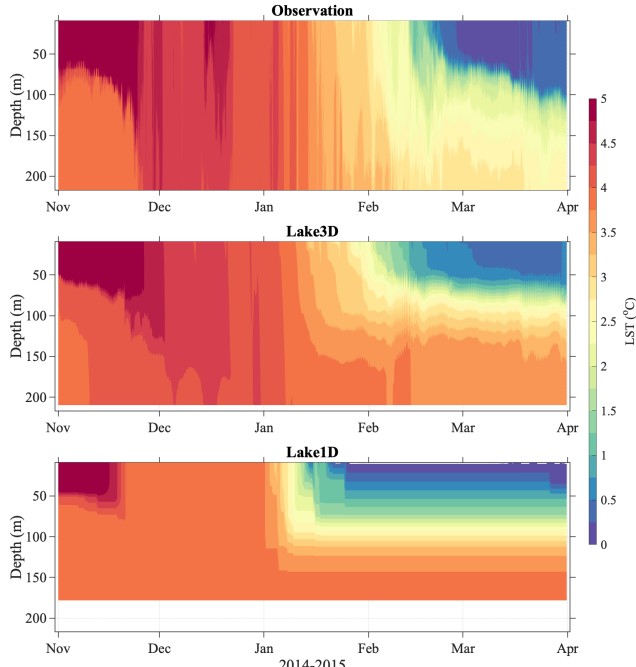


**Figure 4.** Seasonal evolution of daily vertical temperature (°C) profiles from the thermistor observations
(top panel), NU-WRF/FVCOM 3D lake model (middle panel), and NU-WRF/LISSS 1D lake model
(bottom panel) at Spectacle Reef in Lake Huron during November 2014-March 2015.
Correspondingly, NU-WRF/FVCOM resolves the spatiotemporal evolution of lake ice cover
very well across all of the lakes with RMSE of percent ice cover less than 8% for Lakes Huron,
Michigan, and Ontario and 11% and 18% for Lakes Superior and Erie, respectively (Fig. 2
lower panels).  The 3D lake model and GLSEA data exhibit similar seasonal trends both in
timing and magnitude, with ice cover typically starting to rapidly increase in January, peaking
in February and early March, and declining thereafter (Fig. 2). Lake Erie shows the earliest and
sharpest increase in ice cover, peaking near 100% in early February and throughout mid-March,
indicative of its shallower depth and weaker thermal inertia. Lakes Huron and Superior show a




persistent increase in ice cover through February, with peak coverage of >90% occurring at the
beginning of March. Lakes Michigan and Ontario exhibit more gradual increases and lower
peaks in ice cover. The model appears to capture the general seasonal trends of the GLSEA
data with high fidelity, although some discrepancies are evident, particularly over Lakes Erie
and Superior (Fig. 2).
NU-WRF/FVCOM performs reasonably well in mirroring the general spatial patterns of lake
ice cover (Fig. 5, top and middle panels). For January, the GLSEA data shows a pronounced ice
formation in the nearshore regions across the lakes, with the greatest ice concentration visible
along the coastlines and very limited ice cover in offshore waters. The model captures this
nearshore ice development quite well, although it suggests less ice cover in the offshore areas,
particularly over Lake Erie. In February, the extent of ice cover varies dramatically across the
lakes, including nearly full ice cover on Lake Erie and significant ice-free areas on Lake
Ontario, as well as for Lakes Michigan and Huron, which have distinctly less ice cover in their
southern and central basins, respectively. The model captures this variability very well, while
slightly overestimating the ice cover in the central regions of Lake Superior. For March, the
model successfully replicates the patterns of significant declines in ice cover in the western
sections of the lakes, with much higher ice coverage in the eastern sections of the lakes.
On the other hand, NU-WRF/LISSS (Lake1D) generates excessive ice cover during January,
when both observations and NU-WRF/FVCOM suggested that the majority of the lakes were
ice-free. In February, the excessive ice cover simulated by the NU-WRF/LISSS model persists,
with near 100% ice coverage over all of the lakes, and the model fails to depict the large spatial
variability across the lakes. Such a persistent overestimation of ice cover throughout the cold
season by NU-WRF/LISSS was also reported in Notaro et al. (2021).



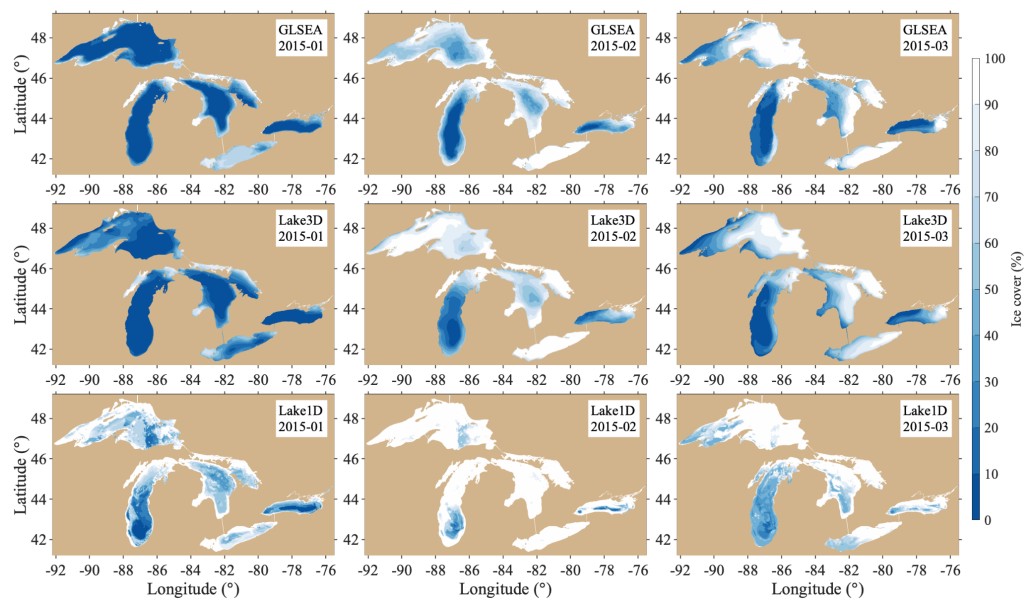


**Figure 5.** Spatial patterns of mean percent lake ice cover from GLSEA data (top panels), NU-WRF/FVCOM 3D lake model simulations (middle panels), and NU-WRF/LISSSS 1D lake model simulations (bottom panels) for January 2015 (first column), February 2015 (second column), and March 2015 (third column).

### 4.2 Over-lake Latent and Sensible Heat Fluxes

The improved LST and ice simulation by the 3D lake model translates to an improvement in the simulated over-lake latent and sensible heat fluxes, particularly for the ice-cover season (Fig. 6). The observations for upward latent and sensible heat fluxes from two eddy covariance flux towers at Granite Island on Lake Superior and Spectacle Reef on Lake Huron are compared against the simulated fluxes from NU-WRF/FVCOM (Lake3D) and NU-WRF/LISSS (Lake1D). The two lakes are selected for demonstration as they have the highest ice coverage during the simulation period. NU-WRF/LISSS reasonably simulates the magnitude and variability of the heat fluxes from November until mid-December, similar to the observations and NU-WRF/FVCOM, although with larger biases. However, it grossly underestimates the fluxes during the ice-cover season (January-March) by simulating a nearly constant near-zero flux. This is mainly due to the excessive ice cover simulated by the 1D lake model, which creates a physical barrier for air-lake energy fluxes. Since the 3D lake model more accurately

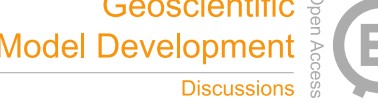

simulates the LST and ice cover, it successfully captures the magnitude and variability of the
heat fluxes, even during the ice-cover season, with RMSEs that are 50% lower than those from
the 1D lake model (Fig. 6). Latent heat in Spectacle Reef is the only exception, where NU-
WRF/FVCOM struggles to capture the magnitude of the upward latent heat flux due to the
overestimated ice cover at the site. However, it still outperforms NU-WRF/LISSS in terms of
capturing the seasonal trend in latent heat fluxes.

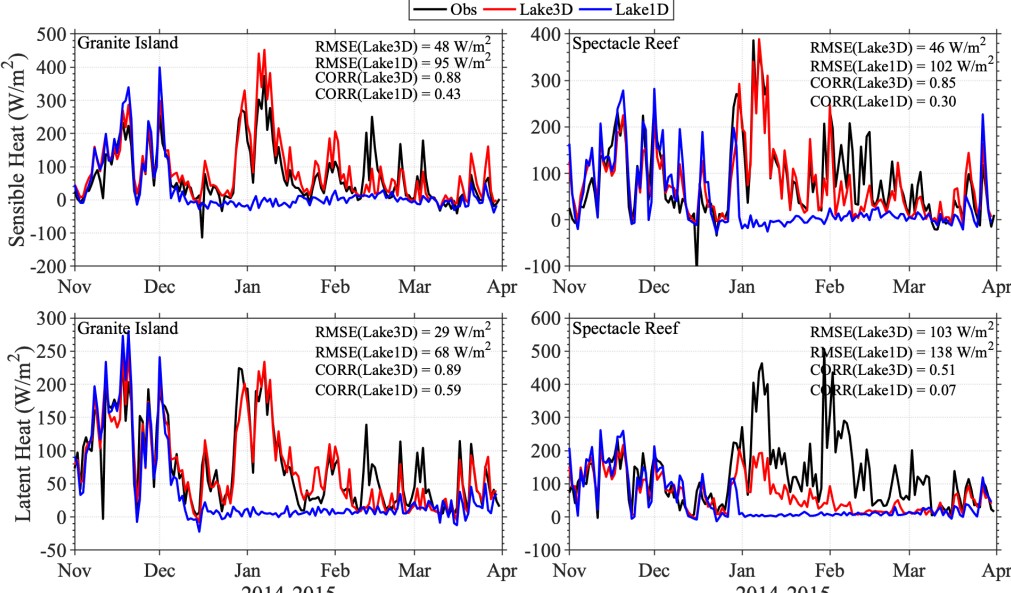


**Figure 6.** Time series of daily sensible (upper panels) and latent (lower panels) heat fluxes (W/m²) from
GLEN observations (black lines), NU-WRF/FVCOM 3D lake model simulations (red lines), and NU-
WRF/LISSS 1D lake model simulations (blue lines) at Granite Island on Lake Superior (left) and
Spectacle Reef on Lake Huron (right). The RMSE and temporal correlations between the simulations and
GLEN observations are provided in each panel.
**4.3    Over-lake Air Temperature and Wind**
Along with the improved simulation of the Great Lakes' physical characteristics and surface
heat fluxes, NU-WRF/FVCOM improves the simulated over-lake atmospheric state across the
Great Lakes, including air temperature and wind speed. The cold air temperature biases
produced over the lakes by NU-WRF/LISSS are significantly reduced (Fig. 7) with better



simulated, more intense upward heat fluxes in January. This improvement in the simulated air
temperature at the two sites, Granite Island and Spectacle Reef, is clearly evident. Similar to the
fluxes, NU-WRF/LISSS modeled air temperature diverges from the observations in January
and February, with a noticeable cold bias. This cold bias is the result of significant suppression
of the upward heat fluxes during those months in the 1D lake model due to excessive simulated
ice cover. NU-WRF/FVCOM, on the other hand, produces a much warmer and more accurate
over-lake air temperature for January and February due to its reasonable representation of
upward heat fluxes. The simulated wind speed over the lakes is also improved, especially in
January-February (Fig. 7). This advancement is attributed to the refined simulation of surface
roughness (i.e., ice versus water), the water-air temperature gradient, and associated instability
over the lakes due to decreased ice cover. Large wind spikes (16 m/s) in January-February are
more accurately captured by NU-WRF/FVCOM.

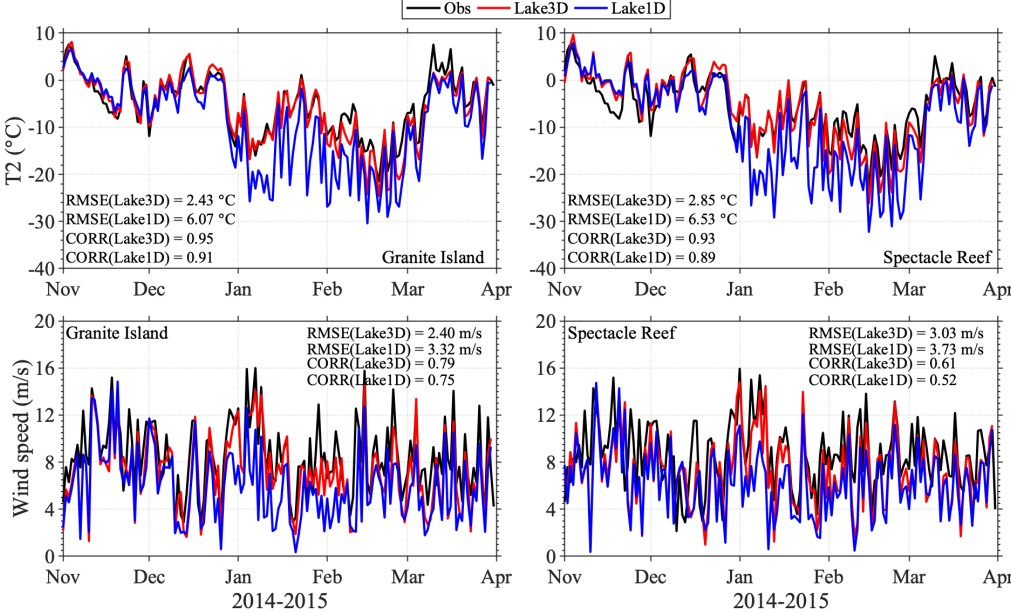


**Figure 7.** Time series of daily air temperature (°C, upper panels) at 2-m height (T2) and wind speed
(m/s, lower panels) from GLEN observations (black lines), NU-WRF/FVCOM 3D lake model
simulations (red lines), and NU-WRF/LISSSS 1D lake model simulations (blue lines) at Granite Island
on Lake Superior and Spectacle Reef on Lake Huron during November 2014-March 2015. The RMSE
and temporal correlations between the simulations and GLEN observations are provided in each panel.



## 5    Discussion

The Great Lakes modeling community has agreed on the pressing need to integrate 3D lake
models instead of conventional 1D lake modeling in the Great Lakes regional climate studies
(Delaney and Milner, 2019). However, no studies have yet detailed the key 3D hydrodynamic
processes that explain the superiority of 3D lake models over 1D lake models, especially
regarding cold season performance and lake-atmosphere interactions. The primary goal of this
study is to identify the crucial processes influencing lake thermal structure and ice cover that
are missed by 1D lake models but effectively captured by 3D lake models, through a series of
process-oriented experiments presented below.

### 5.1    Impact of Ice Movement

The 3D hydrodynamic model, FVCOM, includes an embedded unstructured-grid ice model
capable of resolving several components for atmosphere-ice-water interactions (Gao et al.,
2011). It includes a thermodynamic model that computes the local growth rates of snow and ice
due to vertical conductive, radiative, and turbulent fluxes, aligning with features typically
included in 1D lake models (Bitz and Lipscomb, 1999). More importantly, it features an ice
dynamics model that predicts the ice pack's velocity field based on its material strength; a
transport model that describes the advection of areal concentration, ice volumes, and other state
variables; and a ridging parameterization that facilitates the transfer of ice among thickness
categories (Hunke et al., 2010).
Case C2-1 (NoIceTransp) is designed to examine the impact of ice transport on LSTs and
overlying atmospheric conditions, compared to standard case C1-1 (Lake3D). In case C2-1, ice
dynamics, velocity fields, and ice pack transport are disabled in FVCOM. Instead, only ice
thermal dynamics are simulated, as in the 1D lake model. Figure 8 compares cases C1-1
(Lake3D) and C2-1 (NoIceTransp), illustrating their performance in simulating the observed
spatial pattern of ice coverage in March 2015, characterized by open water on the western side
of the Great Lakes and predominant ice cover on the eastern side (Fig. 8a). Utilizing a 3D lake
model that only accounts for ice thermal dynamics results in an overestimation of ice cover,
with near 100% lakewide ice cover in Lakes Superior, Huron, and Erie (Fig. 8b). However,
integrating ice dynamics, including transport influenced by wind and water-ice stress, results in
excellent agreement with observations, highlighting the critical role of ice transport in accurate
ice modeling (Fig. 8c). This pattern aligns with the modeled ice velocities, which attribute the
eastward ice cover distribution to dominant eastward ice transport (Fig. 8d). Under cold winter
conditions characterized by strong westerly winds, ice is driven eastward, maintaining open
water in the lake's western part. This facilitates ongoing atmospheric interactions, allowing for
heat release. Neglecting these dynamics leads to unrealistic ice accumulation by diminishing
the influence of wind on surface water movement and mixing. This overaccumulation of ice
cover hampers the efficiency of vertical turbulent mixing, which is essential for maintaining a
warmer surface layer, thereby exacerbating ice formation and accumulation. The incorporation
of ice dynamics into 3D lake models is thus essential for accurately simulating ice distribution,
emphasizing the necessity of resolving ice transport to replicate observed patterns accurately.

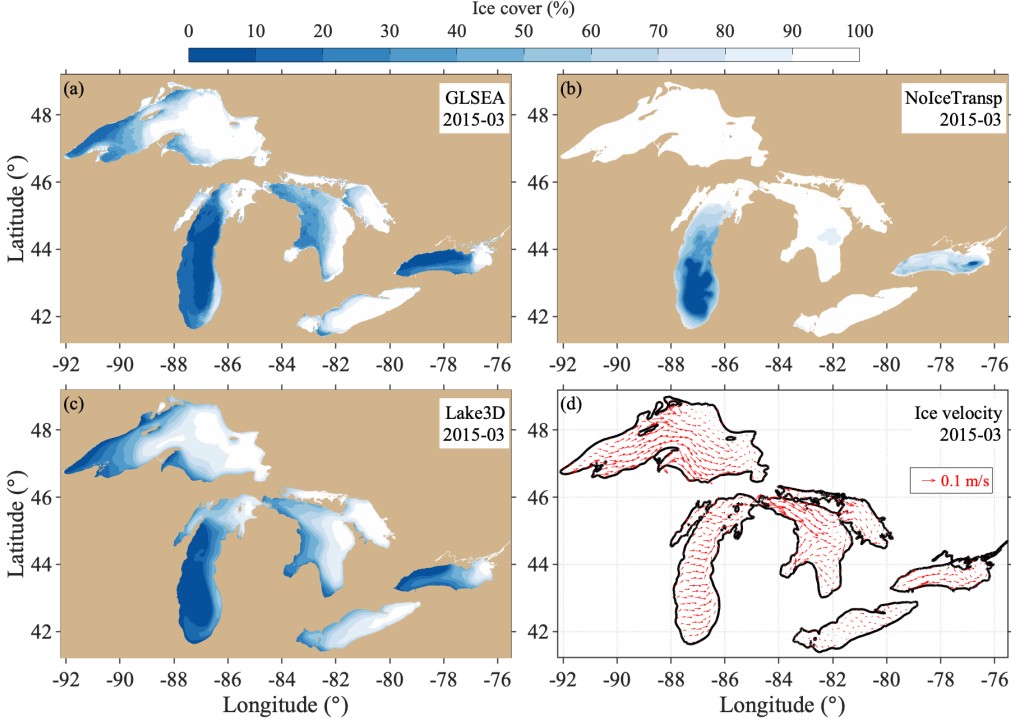

**Figure 8.** Spatial patterns of mean percent lake ice cover from GLSEA data (a), case 2-1 (NoIceTransp)
simulations (b), and case 1-1 (Lake3D) standard simulations (c), along with simulated mean ice
velocities (m/s) during (d) March 2015.



**5.2    Impact of Heat Transport**
The 3D lake model also resolves the advective transport of heat associated with the simulated
circulation. The advective transport and turbulent mixing of temperature in the 3D lake model
are governed by following equation:
$$\frac{\partial T}{\partial t} + u\frac{\partial T}{\partial x} + v\frac{\partial T}{\partial y} + w\frac{\partial T}{\partial z} = \frac{\partial}{\partial z}\left(K_h\frac{\partial T}{\partial z}\right) + F_T \qquad (1)$$

with the surface heat flux boundary condition:
$$\frac{\partial T}{\partial t} = \frac{1}{\rho c_p K_h}[LW(x,y,t) - LH(x,y,t) - SH(x,y,t)] \qquad (2)$$

where $T$ is the water temperature and $u$, $v$, and $w$ are the $x$, $y$, and $z$ components of the water
velocity, respectively. $K_h$ is the vertical thermal diffusivity coefficient and $F_T$ is the horizontal
diffusion term. $\rho$ is water density, $c_p$ is the specific heat capacity of water and
$LW(x,y,t)$, $LH(x,y,t)$, and $SH(x,y,t)$ are net longwave radiation, upward latent heat and
sensible heat fluxes varying in space and time, respectively.
Case C2-2 (NoHeatAdv) analyzes the impact of 3D heat transport. In this case, the 3D
temperature advection terms ($u\frac{\partial T}{\partial x}$, $v\frac{\partial T}{\partial y}$, $w\frac{\partial T}{\partial z}$) are turned off.
Comparing the standard simulation C1-1 (Lake3D) to case C2-2 (NoHeatAdv), Figure 9
demonstrates that, in the absence of advective heat transport by lake currents, the surface
temperatures can remain consistent with the basic patterns observed in the standard 3D lake
simulation throughout the entire simulation period. The differences in the time series of lake-
wide average LSTs for the five lakes are small, with a maximum difference of 0.4°C between
the two cases. The spatial patterns of LST biases, when compared with GLSEA, are generally
more noticeable, with the most significant positive biases (~ 2°C) concentrated around the
coastal waters of the Great Lakes and eastern Lake Erie from January to March 2015 and larger
negative biases (~ 3°C) in the central basin of Lake Huron in November 2014 in the
NoHeatAdv case.



In addition, the vertical transport associated with upwelling, resolved by the 3D model, brings
relatively warmer water from deep in the lake to the surface. This vertical transport mechanism
cannot be represented in 1D lake models that only account for vertical diffusion. This can
create significant local-scale differences along the coast, as shown on the western shore of Lake
Superior in March 2015 [Fig. 9, bottom panels. Notice that the GLSEA is not able to well
capture coastal upwelling (Ye et al., 2020)]. This underscores the importance of including
advective heat transport to accurately resolve the redistribution of heat within the lake. The
inclusion of advective dynamics, by facilitating both lateral and vertical redistribution, enables
a more realistic simulation of the complex spatial heat patterns within large lake systems.

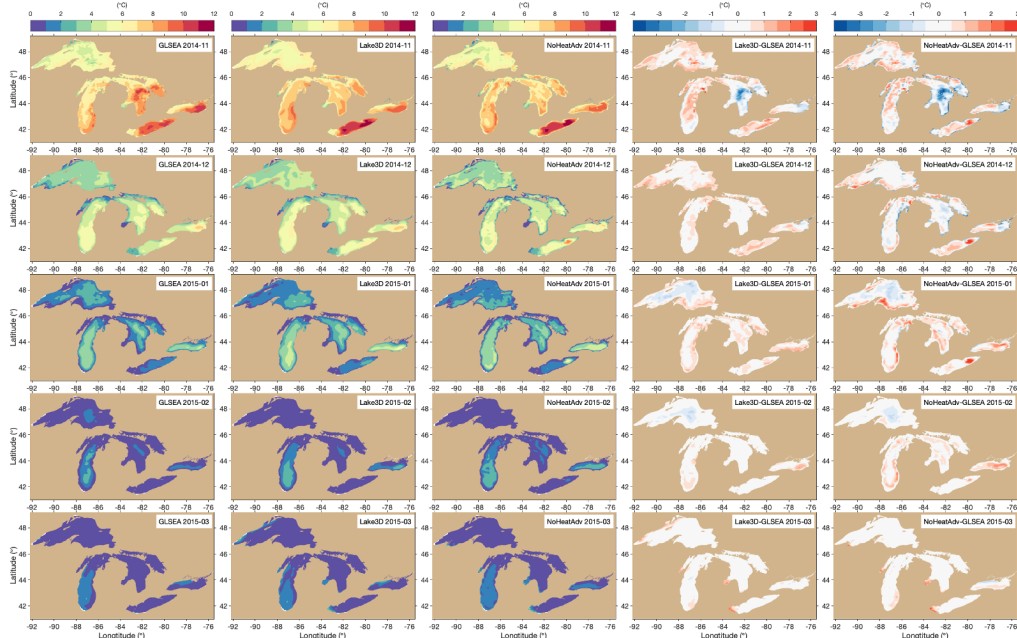

**Figure 9.** Spatial patterns of mean LSTs (°C) from GLSEA data (first column), case C1-1 (Lake3D)
standard simulations (second column), and case C2-2 (NoHeatAdv) simulations (third column) from
November 2014 (top row) to March 2015 (bottom row). Their monthly biases relative to GLSEA data are
presented in the fourth and fifth columns, respectively.
Capturing the evolution of the vertical thermal structure within the deep water is particularly
challenging in lake models. As previously shown in Fig. 4, the in-situ thermistor measurement



at Spectacle Reef on Lake Huron is located in a deep region with a water depth greater than 200
meters. Case C2-2 (NoHeatAdv) generally reproduced the thermal patterns from case C1-1
(Lake3D) in terms of both timing and intensity of summer stratification, fall turnover, and
winter inverse stratification (Fig. 10a,b). While the comparison shows that the overall thermal
structures are similar in both simulations, there is a noticeable difference within the subsurface
layer, specifically between 50 to 100 meters in depth (Fig. 10c), suggesting that heat advection
might have a more significant impact on temperature distribution in the subsurface layer of the
water column in this case. Without accounting for heat advective transport, there appears to be
artifacts of stepwise vertical thermal gradients in case C2-2.

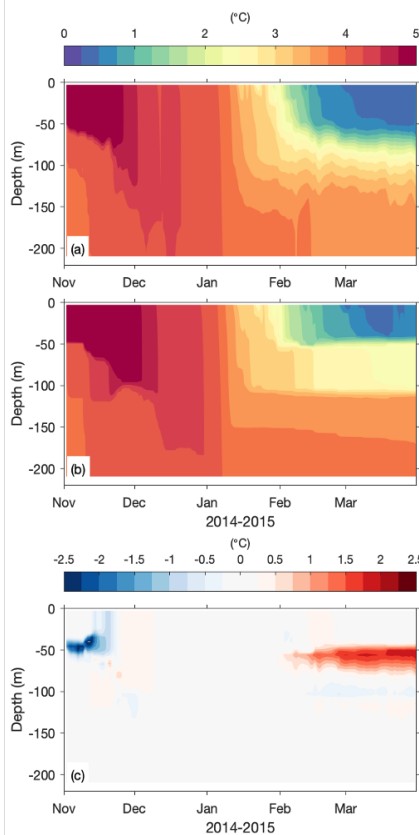


**Figure 10.** Mean vertical temperature (°C) profiles from a): case C1-1 (Lake3D) standard run and b):
case C2-2 (NoHeatAdv) and c): their difference at Spectacle Reef in Lake Huron during November 2014-
March 2015.



To gain a deeper understanding of the results, we analyzed the heat balance to identify the
contributions of different physical processes. This analysis involved examining each term in the
temperature governing equation (Eq. 1) that is directly computed in FVCOM over the
simulation period. The temperature change is driven by 3D advective heat transport, horizontal
heat diffusion, and vertical diffusion due to turbulent mixing.

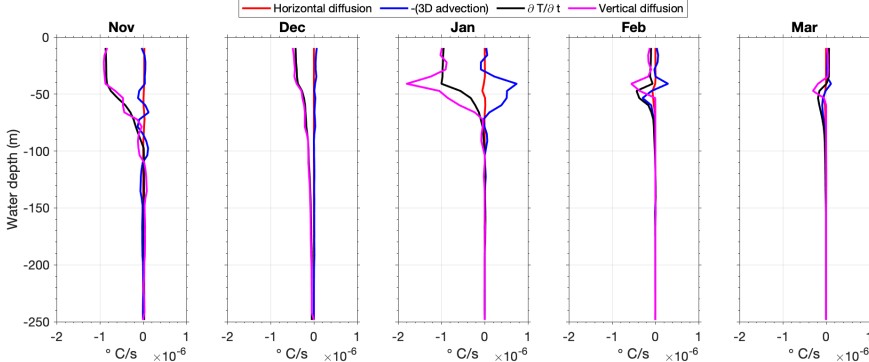

**Figure 11.** Monthly averaged vertical profile of each term of the temperature equation in the C1-1
(Lake3D) simulation from November 2014 to March 2015, output from location at Spectacle Reef in
Lake Huron. The temperature change rate ($\frac{\partial T}{\partial t}$) is determined by 3D advection (blue), horizontal diffusion
(red), and vertical diffusion (purple).

The analysis revealed the relative impact of physical processes on thermal changes during the
winter months (Fig. 11). Significant cooling and decreasing temperatures were observed within
the upper 100 meters of the water column, as indicated by the negative temperature change over
time ($\frac{\partial T}{\partial t}$) in this zone. In contrast, the water below 100 meters in depth remained largely
unchanged and stable in temperature. In November and December, vertical turbulent mixing
processes primarily controlled the cooling rate in the upper 100 meters, during which surface
heat fluxes served as net losses from the lake along with vigorous turbulent mixing in the lake.
Advection played a much less important role in temperature changes during this period.
However, starting in January, 3D advection played an important role in redistributing heat in
the 25-100 meter layer, offsetting some of the cooling induced by surface heat loss through
mixing. In February and March 2015, advection proved to be significant at the lower boundary



of the surface mixed layer. These observations explain the larger temperature difference in the
subsurface layer between cases C1-1 (Lake3D) and C2-2 (NoHeatAdv) (Fig. 10), highlighting
the evolving balance between vertical diffusion and advection in controlling the epilimnetic
heat budget and temperature changes in large lakes during the cold season.
**5.3    Impact of Vertical Mixing**
The analysis above (Fig. 11) highlights the dominant factor, vertical turbulent mixing, in
determining seasonal lake temperature change. Note that we have already discussed the
importance of ice transport associated with currents as well as the impact of advective heat
transport. To understand the mechanism responsible for the differing performance between the
1D and 3D lake models in simulating vertical mixing, we examine how vertical turbulent
mixing is calculated in these two types of models. The intensity of vertical mixing in both
models is represented by vertical eddy diffusivity, which is determined by turbulent kinetic
energy ($q^2$). In the 3D hydrodynamic lake model, a sophisticated 3D turbulence closure model
is used, in which a prognostic equation predicts the change rate of $q^2$ based on its advection,
and its turbulence production, including both shear-induced production ($P_s$) and buoyancy-
induced production ($P_b$), and its dissipation rate ($\epsilon$), as well as its diffusion. This equation is
complemented by either a separate prognostic equation for dissipation rate (k-$\epsilon$; Launder and
Spalding, 1974) or a diagnostic equation for turbulent mixing length (Meller and Yamada,

638    1982).

The equation governing the evolution of turbulent kinetic energy ($q^2$) is
$$\frac{\partial q^2}{\partial t} + u\frac{\partial q^2}{\partial x} + v\frac{\partial q^2}{\partial y} + w\frac{\partial q^2}{\partial z} = P_s + P_b - \epsilon + \frac{\partial}{\partial z}\left(K_q \frac{\partial q^2}{\partial z}\right) + F_q \qquad (3)$$
where $q^2 = (\langle u'^2 \rangle + \langle v'^2 \rangle + \langle w'^2 \rangle)/2$, with $u', v', w'$ represent the fluctuating components of
velocity in the $x, y, z$ directions, respectively. The $\langle\ \rangle$ denotes averaging over time or space to
obtain the mean. Shear production is often approximated as $P_s = K_m((\frac{\partial u}{\partial z})^2 + (\frac{\partial v}{\partial z})^2)$, where
$K_m$ is the vertical eddy viscosity coefficient. Buoyancy production is computed as $P_b =$
$-\frac{g}{\rho_0} K_h \frac{\partial \rho}{\partial z}$, where $g$ is acceleration due to gravity. $\rho_0$ is reference density of the fluid (e.g.,



ocean water or air). $K_h$ is thermal diffusivity, $\frac{\partial \rho}{\partial z}$ is vertical gradient of density, indicating
stratification. The turbulent kinetic energy dissipation rate is represented as $\epsilon = q^3/Bl$ , where
$l$ is the turbulence length scale and B is an empirical constant. $K_{m,h,q} = qlS_{m,h,q}$, where $S_{m,h,q}$
are stability functions for $K_{m,h,q}$, respectively. $K_q$ is the vertical diffusivity coefficient for
turbulent kinetic energy and $F_q$ is horizontal diffusion of the turbulent kinetic energy .
Figure 12 reveals that in the Great Lakes, shear production—induced by the vertical gradient of
horizontal velocity in the water column—is the primary driver of subsurface turbulent mixing.
Conversely, buoyancy production plays a secondary role, being at least one order of magnitude
smaller than shear production in the first 50 meters of depth. This underscores the importance
of including accurate current simulation when estimating the vertical turbulent mixing, which is
crucial for accurately simulating heat exchange in the water column and ultimately determining
the lake's thermal structure and ice formation.

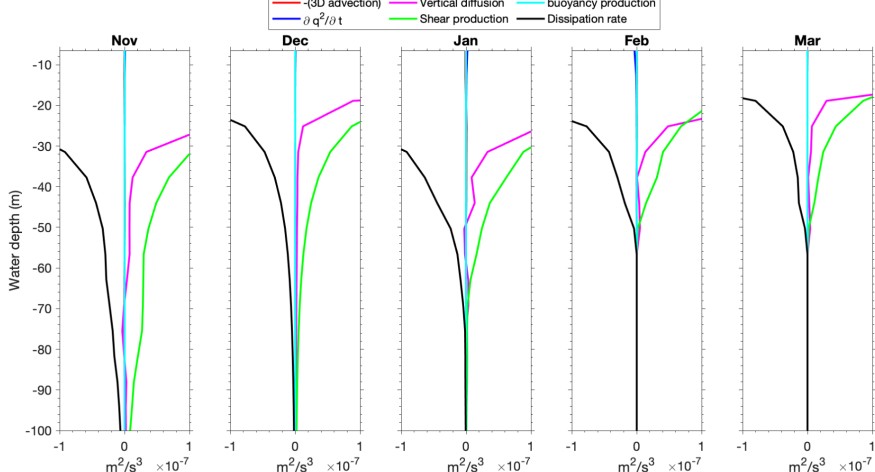


**Figure 12.** Monthly averaged vertical profile of each term of the turbulence kinetic equation in the C1-1
(Lake3D) simulation from November 2014 to March 2015, at Spectacle Reef on Lake Huron. The change
rate of turbulent kinetic energy (blue) is based on the 3D advection (red), and the turbulence production,
including both shear-induced production (green) and buoyancy-induced production (cyan), the
dissipation rate (black), and vertical diffusion (purple).



Figure 13 compares the vertical temperature profiles between the standard simulation C1-1
(Lake3D) and case C2-3 (NoShearProd). The NoShearProd case shows much stronger
stratification, particularly from January to March. The absence of shear production leads to
significantly reduced turbulent mixing and limiting heat exchange between surface and deeper
waters, which results in a much colder surface layer (0-40 m) in January and much warmer
deep waters (50-150 m) in February and March compared to the standard run. Consequently,
the colder surface water temperature favors ice formation, leading to overestimated ice cover in
the NoShearProd case compared to the standard simulation and observations, particularly in
January and February (Fig. 14).

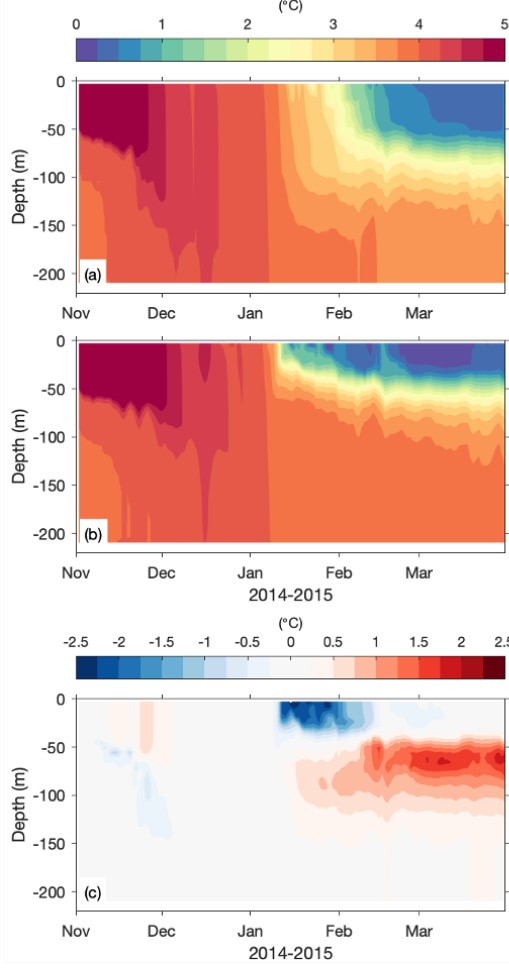




**Figure 13.** Mean vertical temperature (°C) profiles from a): case C1-1 (Lake3D) standard run and b): case C2-3 (NoShearProd) and c): their difference at Spectacle Reef on Lake Huron during November 2014-March 2015.

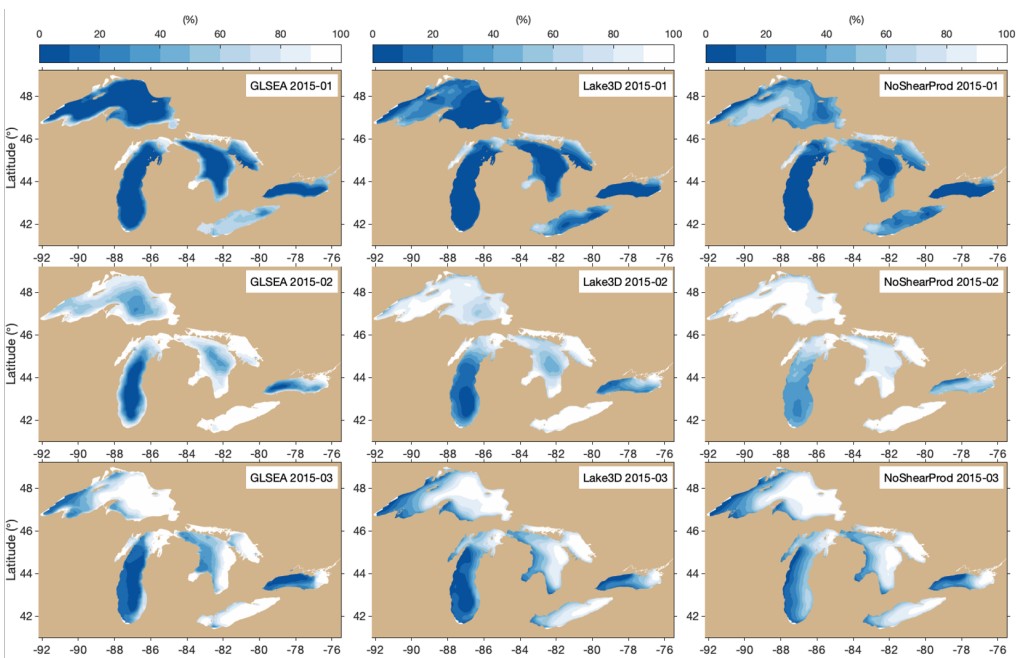

**Figure 14.** Spatial patterns of mean percent lake ice cover from GLSEA data (first column), case C1-1 (Lake3D; second column) and case 2-3 (NoShearProd; third column) for January 2015 (first row), February 2015 (second row), and March 2015 (third row).

LISSS, as true of other 1D lake models, was originally designed for small and shallow inland lakes and was not designed to resolve water currents (Subin et al., 2012; Notaro et al., 2021). Some other 1D lake models (Stepanenko and Lykossov, 2005; Stepanenko et al., 2011) employ a crude representation of average flow fields. Therefore, 1D lake models rely on empirical or semi-empirical relationships to estimate how wind stress affects the lake's turbulence and mixing without explicitly resolving 3D velocity fields. These thermal diffusion-based models often employ a latitude-dependent Ekman decay, accompanied by an empirical modification factor, to estimate a lumped eddy diffusivity coefficient as an approximation for surface wind-induced mixing (Xiao et al., 2016). Thus, the lack of accurate simulation of turbulent mixing





processes makes the 1D model of limited capacity in accurately simulating the Great Lakes'
thermal structure.

## 6   Summary and Conclusion

In summary, a two-way coupled NU-WRF/FVCOM model (CLIAv1) has been developed
toward the next generation of a regional climate model for the Great Lakes Basin for accurate
representations of lake–ice–atmosphere interactions. NU-WRF/FVCOM significantly improved
on the performance of NU-WRF coupled with an optimized 1D lake model, and accurately
reproduced the physical characteristics of the Great Lakes (e.g., LST, ice cover, and thermal
structure).  This led to further improvements in simulated over-lake atmospheric conditions
(e.g., air temperature, wind, latent and sensible heat) through two-way lake-atmosphere
interactions.
While 1D column lake models have been widely used in the simulations of inland lakes
worldwide, small inland lakes and the Great Lakes exhibit fundamental differences in their
physical characteristics, such as size and depth, which in turn influence their mixing behaviors,
thermal structures, and circulation patterns. Inland lakes, generally much smaller (with a typical
average area of 1-10 kilometers) and much shallower (with a typical average depth of ~10m),
respond more rapidly to atmospheric conditions. This leads to a fairly uniform horizontal
pattern and a simpler mixing process in response to surface wind, due to their shallow depth
and small thermal inertia. Therefore, 1D column lake models serve as an appropriate and
efficient tool for simulating inland lake processes, particularly when the lake depth is shallower
than 20 meters. In contrast, the vast size (e.g., Lake Superior alone covers about 82,100 square
kilometers) and significant depth (e.g., the average depth of Lake Superior is 147 m, with a
maximum depth of 400 m) of the Great Lakes result in complex hydrodynamic and thermal
dynamics. This complexity causes the Great Lakes to exhibit many sea-like characteristics.
This study has highlighted key physical processes that differentiate the large, deep Great Lakes
from small, shallow inland lakes, and how these processes impact lake simulations.
Specifically, we identified that ice dynamics, particularly ice transport, are vital in the Great
Lakes, influencing ice cover formation and heat exchange between the lake and the atmosphere.





Secondly, we show that advective heat transport, facilitates both lateral and vertical
redistribution, enables a more realistic simulation of the complex spatial temperature patterns,
particularly the predominance of advective heat transport in the subsurface layers. Thirdly, we
identified the critical role of resolving shear production in turbulent mixing in the Great Lakes,
which is the most influential factor that determines heat transfer and, subsequentially, lake
thermal structure. Ice transport, heat transfer, and shear production in turbulence mixing are
fundamentally linked to the 3D lake currents, which are missing or crudely represented in 1D
lake models. Our findings underscore that circulation currents are pivotal in the physical
limnology of the Great Lakes. Given the ongoing impact of climate change on these aquatic
systems (Zhong et al., 2016; Woolway et al., 2021; Cannon et al., 2024), accurately
incorporating 3D lake dynamics becomes crucial for projecting future thermal structures and
ecosystem effects.
Lastly, we acknowledge that there are multiple ways to tune the 1D lake column model or build
an accurate empirical relationship between atmospheric conditions and the strength of mixing
to improve 1D model simulations. However, the major challenge with this approach is that any
empirical or simplified physical relationship carries significant risks of not holding in the
future, especially in the context of climate change. While it may work well to calibrate the
model based on a substantial amount of validation data, this approach has a much larger risk
and lacks reliability if the model is used for climate projections where conditions change
significantly. Therefore, we advocate for the complete integration of 3D hydrodynamic lake
models in a two-way coupled fashion to project future changes in large freshwater systems.
This method ensures that projections are based on physical processes, reducing the risk
associated with empirical relationships and increasing the model's reliability for future climate
scenarios.
**Code and data availability**
The source codes of CLIAv1 with the two-way coupled FVCOM and NU-WRF used in this
study are available at https://doi.org/10.5281/zenodo.12746348 (Huang, 2024a) and
https://doi.org/10.5281/zenodo.12746306 (Huang, 2024b) respectively. The GLSEA data were
obtained from the NOAA Coastwatch website (https://coastwatch.glerl.noaa.gov/glsea/doc/)



(GLSEA, 2023). The GLEN data were from the Lake Superior Watershed Partnership website
(https://superiorwatersheds.org/GLEN/), with data compilation and publication provided by
LimnoTech under Award/Contract 10042-400759 from the International Joint Commission
(IJC) through a subcontract with the Great Lakes Observing System (GLOS).

**Author contributions**

PX conceived the study. PX and CH developed the model code. PX designed the experiments.
PX, MN, XZ, CH, MBK, and CZ conducted the analyses. PX, MN, MBK, and CZ wrote the
original manuscript. All others contributed to revising the manuscript. All authors have read
and agreed to the published version of the manuscript.

**Competing interests**

The authors declare that they have no conflict of interest.

**Acknowledgments**

This is the Contribution No. 121 of the Great Lakes Research Center at Michigan
Technological University. The study was funded by NASA's Modeling, Analysis, and
Prediction Program (Grant 80NSSC17K0287 and Grant 80NSSC17K0291). Hydrodynamic
modeling was also partly supported by the U.S. Department of Energy, Office of Science, under
award number DE-SC0024446. The statements, findings, conclusions, and recommendations of
authors expressed herein do not necessarily state or reflect those of the United States
Government or any agency thereof.





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

B. Macelwane Annual Award in Meteorology, announced at the Annual Meeting of the
AMS, Philadelphia, Pa., 21 January 1976. *Bulletin of the American Meteorological*
*Society, 57*(5), 548-553.
Craig, A., Valcke, S., & Coquart, L. (2017). Development and performance of a new version of
the OASIS coupler, OASIS3-MCT_3. 0. *Geoscientific Model Development, 10*(9),
3297-3308.
Crossman, E. J., & Cudmore, B. C. (1998). Biodiversity of the fishes of the Laurentian Great
Lakes: a great lakes fishery commission project. *Italian Journal of Zoology, 65*(S1),
357-361.
Delaney, F., & Milner, G. (2019). *The State of Climate Modeling in the Great Lakes Basin - A*
*Synthesis in Support of a Workshop held on June 27, 2019 in Arr Arbor, MI*. Retrieved
from https://climateconnections.ca/app/uploads/2020/05/The-State-of-Climate-
Modeling-in-the-Great-Lakes-Basin_Sept132019.pdf
Eichenlaub, V. L. (1978). Weather and climate of the Great Lakes region [USA]. *University of*
*Notre Dame Press*.
Environmental Protection Agency (EPA). (2014). State of the Great Lakes 2011. *EPA 950-R-*
*13-002*. Retrieved from https://archive.epa.gov/solec/web/pdf/sogl-2011-technical-
report-en.pdf
Gao, G., Chen, C., Qi, J., & Beardsley, R. C. (2011). An unstructured-grid, finite-volume sea
ice model: Development, validation, and application. *Journal of Geophysical Research:*
*Oceans, 116*(C8).
Gao, Y., Fu, J. S., Drake, J., Liu, Y., & Lamarque, J.-F. (2012). Projected changes of extreme
weather events in the eastern United States based on a high resolution climate modeling
system. *Environmental Research Letters, 7*(4), 044025.
Gerbush, M. R., Kristovich, D. A., & Laird, N. F. (2008). Mesoscale boundary layer and heat
flux variations over pack ice–covered Lake Erie. *Journal of Applied Meteorology and*
*Climatology, 47*(2), 668-682.
Giorgi, F., & Gutowski Jr., W. J. (2015). Regional Dynamical Downscaling and the CORDEX
Initiative. *Annual Review of Environment and Resources, 40*(1), 467-490.
doi:10.1146/annurev-environ-102014-021217
Goudsmit, G. H., Burchard, H., Peeters, F., & Wüest, A. (2002). Application of k-ε turbulence
models to enclosed basins: The role of internal seiches. *Journal of Geophysical*
*Research: Oceans, 107*(C12), 23-21-23-13.
Gu, H., Jin, J., Wu, Y., Ek, M. B., & Subin, Z. M. (2015). Calibration and validation of lake
surface temperature simulations with the coupled WRF-lake model. *Climatic Change,*
*129*, 471-483.
Hanrahan, J., Langlois, J., Cornell, L., Huang, H., Winter, J. M., Clemins, P. J., . . . Bruyère, C.
(2021). Examining the Impacts of Great Lakes Temperature Perturbations on Simulated
Precipitation in the Northeastern United States. *Journal of Applied Meteorology and*
*Climatology, 60*(7), 935-949.
Holman, K. D., Gronewold, A., Notaro, M., & Zarrin, A. (2012). Improving historical
precipitation estimates over the Lake Superior basin. *Geophysical Research Letters,*
*39*(3).




Hostetler, S. W., & Bartlein, P. J. (1990). Simulation of lake evaporation with application to
modeling lake level variations of Harney-Malheur Lake, Oregon. *Water Resources*
*Research, 26*(10), 2603-2612. doi:10.1029/WR026i010p02603
Huang, C. (2024a). Lake model code for the manuscript "On the Importance of Coupling a 3D
Hydrodynamic Model with a Regional Climate Model in Simulating the Great Lakes
Winter Climate" [Software]. Zenodo. doi:https://doi.org/10.5281/zenodo.12746348
Huang, C. (2024b). NU-WRF (v11) code for the manuscript "On the Importance of Coupling a
3D Hydrodynamic Model with a Regional Climate Model in Simulating the Great
Lakes Winter Climate" [Software]. Zenodo.
doi:https://doi.org/10.5281/zenodo.12746306
Hunke, E. C., & Dukowicz, J. K. (1997). An elastic–viscous–plastic model for sea ice
dynamics. *Journal of Physical Oceanography, 27*(9), 1849-1867.
Hunke, E. C., Lipscomb, W. H., Turner, A. K., Jeffery, N., & Elliott, S. (2010). Cice: the los
alamos sea ice model documentation and software user's manual version 4.1 la-cc-06-
012. *T-3 Fluid Dynamics Group, Los Alamos National Laboratory, 675*, 500.
Hutson, A., Fujisaki-Manome, A., & Lofgren, B. (2024). Testing the Sensitivity of a WRF-
based Great Lakes Regional Climate Model to Cumulus Parameterization and Spectral
Nudging. *Journal of Hydrometeorology*. doi:https://doi.org/10.1175/JHM-D-22-0234.1
Kain, J. S. (2004). The Kain–Fritsch convective parameterization: an update. *Journal of*
*Applied Meteorology, 43*(1), 170-181.
Kain, J. S., & Fritsch, J. M. (1990). A one-dimensional entraining/detraining plume model and
its application in convective parameterization. *Journal of Atmospheric Sciences,*
*47*(23), 2784-2802.
Kayastha, M. B., Huang, C., Wang, J., Pringle, W. J., Chakraborty, T., Yang, Z., . . . Xue, P.
(2023). Insights on Simulating Summer Warming of the Great Lakes: Understanding
the Behavior of a Newly Developed Coupled Lake-Atmosphere Modeling System.
*Journal of Advances in Modeling Earth Systems, 15*(7), e2023MS003620.
Kristovich, D. A. R., & Laird, N. F. (1998). Observations of Widespread Lake-Effect
Cloudiness: Influences of Lake Surface Temperature and Upwind Conditions. *Weather*
*and Forecasting, 13*(3), 811-821. doi:10.1175/1520-
0434(1998)013<0811:Oowlec>2.0.Co;2
Kumar, S. V., Peters-Lidard, C. D., Tian, Y., Houser, P. R., Geiger, J., Olden, S., . . . Dirmeyer,
P. (2006). Land information system: An interoperable framework for high resolution
land surface modeling. *Environmental Modelling & Software, 21*(10), 1402-1415.
Lam, D. C., & Schertzer, W. M. (1999). *Potential climate change effects on Great Lakes*
*hydrodynamics and water quality*: ASCE Publications.
Lenters, J., Anderton, J., Blanken, P., Spence, C., & Suyker, A. (2013). Assessing the Impacts
of Climate Variability and Change on Great Lakes Evaporation. 2011 Project Reports.
D. Brown, D. Bidwell, and L. Briley, eds. Available from the Great Lakes Integrated
Sciences and Assessments (GLISA) Center. In.
Lofgren, B. M. (2014). Simulation of atmospheric and lake conditions in the Laurentian Great
Lakes region using the Coupled Hydrosphere-Atmosphere Research Model (CHARM).
Mallard, M., Nolte, C., Spero, T., Bullock, O., Alapaty, K., Herwehe, J., . . . Bowden, J. (2015).
Technical challenges and solutions in representing lakes when using WRF in
downscaling applications. *Geoscientific Model Development, 8*(4), 1085-1096.
Mallard, M. S., Nolte, C. G., Bullock, O. R., Spero, T. L., & Gula, J. (2014). Using a coupled
lake model with WRF for dynamical downscaling. *Journal of Geophysical Research:*
*Atmospheres, 119*(12), 7193-7208.



Martynov, A., Sushama, L., & Laprise, R. (2010). Simulation of temperate freezing lakes by
one-dimensional lake models: performance assessment for interactive coupling with
regional climate models. *Boreal environment research, 15*(2), 143.
Martynov, A., Sushama, L., Laprise, R., Winger, K., & Dugas, B. (2012). Interactive lakes in
the Canadian Regional Climate Model, version 5: the role of lakes in the regional
climate of North America. *Tellus A: Dynamic Meteorology and Oceanography, 64*(1),
16226.
Matsui, T., Iguchi, T., Li, X., Han, M., Tao, W.-K., Petersen, W., . . . Kummerow, C. D. (2013).
GPM satellite simulator over ground validation sites. *Bulletin of the American
Meteorological Society, 94*(11), 1653-1660.
Matsui, T., Santanello, J., Shi, J., Tao, W. K., Wu, D., Peters-Lidard, C., . . . Sekiguchi, M.
(2014). Introducing multisensor satellite radiance-based evaluation for regional Earth
system modeling. *Journal of Geophysical Research: Atmospheres, 119*(13), 8450-8475.
Mellor, G. L., & Yamada, T. (1982). Development of a turbulence closure model for
geophysical fluid problems. *Reviews of Geophysics, 20*(4), 851-875.
doi:https://doi.org/10.1029/RG020i004p00851
Minallah, S., & Steiner, A. L. (2021). The effects of lake representation on the regional
hydroclimate in the ECMWF reanalyses. *Monthly Weather Review, 149*(6), 1747-1766.
Mironov, D., Heise, E., Kourzeneva, E., Ritter, B., Schneider, N., & Terzhevik, A. (2010).
Implementation of the lake parameterisation scheme FLake into the numerical weather
prediction model COSMO.
Mitchell, K. (2005). The community Noah land-surface model (LSM). *User's Guide Public
Release Version, 2*(1).
Mlawer, E. J., Taubman, S. J., Brown, P. D., Iacono, M. J., & Clough, S. A. (1997). Radiative
transfer for inhomogeneous atmospheres: RRTM, a validated correlated-k model for
the longwave. *Journal of Geophysical Research: Atmospheres, 102*(D14), 16663-
16682. doi:https://doi.org/10.1029/97JD00237
Mooney, P., Mulligan, F., & Fealy, R. (2013). Evaluation of the sensitivity of the weather
research and forecasting model to parameterization schemes for regional climates of
Europe over the period 1990–95. *Journal of Climate, 26*(3), 1002-1017.
Morrison, H., Thompson, G., & Tatarskii, V. (2009). Impact of cloud microphysics on the
development of trailing stratiform precipitation in a simulated squall line: Comparison
of one-and two-moment schemes. *Monthly Weather Review, 137*(3), 991-1007.
Moukomla, S., & Blanken, P. D. (2017). The estimation of the North American Great Lakes
turbulent fluxes using satellite remote sensing and MERRA reanalysis data. *Remote
Sens., 9*, 141. doi:https://doi.org/10.3390/rs9020141
Nakanish, M. (2001). Improvement of the Mellor–Yamada turbulence closure model based on
large-eddy simulation data. *Boundary-Layer Meteorology, 99*, 349-378.
Nakanishi, M., & Niino, H. (2006). An improved Mellor–Yamada level-3 model: Its numerical
stability and application to a regional prediction of advection fog. *Boundary-Layer
Meteorology, 119*, 397-407.
Nakanishi, M., & Niino, H. (2009). Development of an improved turbulence closure model for
the atmospheric boundary layer. *Journal of the Meteorological Society of Japan. Ser.
II, 87*(5), 895-912.
Niziol, T. A., Snyder, W. R., & Waldstreicher, J. S. (1995). Winter weather forecasting
throughout the eastern United States. Part IV: Lake effect snow. *Weather and
Forecasting, 10*(1), 61-77.
NOAA Great Lakes Surface Environmental Analysis (GLSEA). (2023). Sea Surface
Temperature (SST) from Great Lakes Surface Environmental Analysis (GLSEA)
[Dataset]. [Available from:
https://coastwatch.glerl.noaa.gov/erddap/files/GLSEA_GCS/, accessed  2023/11/09]
Notaro, M., Bennington, V., & Vavrus, S. (2015). Dynamically Downscaled Projections of
Lake-Effect Snow in the Great Lakes Basin* ,+. *Journal of Climate, 28*, 1661-1684.
doi:https://doi.org/10.1175/JCLI-D-14-00467.1
Notaro, M., Holman, K., Zarrin, A., Fluck, E., Vavrus, S., & Bennington, V. (2013a). Influence
of the Laurentian Great Lakes on Regional Climate. *Journal of Climate, 26*(3), 789-
804. doi:https://doi.org/10.1175/jcli-d-12-00140.1
Notaro, M., Zarrin, A., Vavrus, S., & Bennington, V. (2013b). Simulation of Heavy Lake-
Effect Snowstorms across the Great Lakes Basin by RegCM4: Synoptic Climatology
and Variability*,+. *Monthly Weather Review, 141*(6), 1990-2014. doi:10.1175/mwr-d-
11-00369.1
Notaro, M., Zhong, Y., Xue, P., Peters-Lidard, C., Cruz, C., Kemp, E., . . . Vavrus, S. J. (2021).
Cold Season Performance of the NU-WRF Regional Climate Model in the Great Lakes
Region. *Journal of Hydrometeorology, 22*(9), 2423-2454.
doi:https://doi.org/10.1175/JHM-D-21-0025.1
Oleson, K., Lawrence, D., & Bonan, G. B. (2013). Technical description of version 4.5 of the
Community Land Model (CLM). Ncar Tech. Note NCAR/TN-503+STR. National
Center for Atmospheric Research, Boulder.
Perroud, M., Goyette, S., Martynov, A., Beniston, M., & Annevillec, O. (2009). Simulation of
multiannual thermal profiles in deep Lake Geneva: A comparison of one-dimensional
lake models. *Limnology and Oceanography, 54*(5), 1574-1594.
Peters-Lidard, C. D., Houser, P. R., Tian, Y., Kumar, S. V., Geiger, J., Olden, S., . . . Adams, J.
(2007). High-performance Earth system modeling with NASA/GSFC's Land
Information System. *Innovations in Systems and Software Engineering, 3*, 157-165.
Peters-Lidard, C. D., Kemp, E. M., Matsui, T., Santanello Jr, J. A., Kumar, S. V., Jacob, J. P., .
. . Hou, A. (2015). Integrated modeling of aerosol, cloud, precipitation and land
processes at satellite-resolved scales. *Environmental Modelling & Software, 67*, 149-
159.
Petterssen, S., & Calabrese, P. A. (1959). On some weather influences due to warming of the
air by the Great Lakes in winter. *Journal of Atmospheric Sciences, 16*(6), 646-652.
Rau, E., Vaccaro, L., Riseng, C., & Read, J. G. (2020). The Dynamic Great Lakes Economy
Employment Trends from 2009 to 2018. Retrieved from
https://repository.library.noaa.gov/view/noaa/38612
Riley, M. J., & Stefan, H. G. (1988). MINLAKE: A dynamic lake water quality simulation
model. *Ecological Modelling, 43*(3-4), 155-182.
Schwab, D. J., Leshkevich, G. A., & Muhr, G. C. (1999). Automated Mapping of Surface
Water Temperature in the Great Lakes. *Journal of Great Lakes Research, 25*(3), 468-
481. doi:https://doi.org/10.1016/S0380-1330(99)70755-0
Scott, R. W., & Huff, F. A. (1996). Impacts of the Great Lakes on Regional Climate
Conditions. *Journal of Great Lakes Research, 22*(4), 845-863.
doi:https://doi.org/10.1016/S0380-1330(96)71006-7
Sharma, A., Hamlet, A. F., Fernando, H. J. S., Catlett, C. E., Horton, D. E., Kotamarthi, V. R., .
. . Wuebbles, D. J. (2018). The Need for an Integrated Land-Lake-Atmosphere
Modeling System, Exemplified by North America's Great Lakes Region. *Earth's
Future, 6*(10), 1366-1379. doi:https://doi.org/10.1029/2018ef000870
Shi, J., Matsui, T., Tao, W. K., Tan, Q., Peters-Lidard, C., Chin, M., . . . Kemp, E. (2014).



Implementation of an aerosol–cloud-microphysics–radiation coupling into the NASA
unified WRF: Simulation results for the 6–7 August 2006 AMMA special observing
period. *Quarterly Journal of the Royal Meteorological Society, 140*(684), 2158-2175.
Shi, Q., & Xue, P. (2019). Impact of Lake Surface Temperature Variations on Lake Effect
Snow Over the Great Lakes Region. *Journal of Geophysical Research: Atmospheres,*
*124*(23), 12553-12567. doi:10.1029/2019jd031261
Smagorinsky, J. (1963). General Circulation Experiments with the Primitive Equations: I. The
Basic Experiment *Monthly Weather Review, 91*(3), 99-164. doi:10.1175/1520-
0493(1963)091<0099:Gcewtp>2.3.Co;2
Song, Y., Semazzi, F. H., Xie, L., & Ogallo, L. J. (2004). A coupled regional climate model for
the Lake Victoria basin of East Africa. *International Journal of Climatology, 24*(1), 57-
75.
Spence, C., Blanken, P. D., Hedstrom, N., Fortin, V., & Wilson, H. (2011). Evaporation from
Lake Superior: 2. Spatial distribution and variability. *J. Great Lakes Res., 37*, 717-724.
doi:https://doi.org/10.1016/j.jglr.2011.08.013
Spence, C., Blanken, P. D., Lenters, J. D., & Hedstrom, N. (2013). The importance of spring
and autumn atmospheric conditions for the evaporation regime of Lake Superior. *J.*
*Hydrometeor., 14*, 1647-1658. doi:https://doi.org/10.1175/JHM-D-12-0170.1
Spence, C., Hedstrom, N., Blanken, P., Lenters, J., & Cutrell, G. (2019). Great Lakes
Evaporation Network (GLEN) data. Great Lakes Observing System (GLOS). In.
Spero, T. L., Nolte, C. G., Bowden, J. H., Mallard, M. S., & Herwehe, J. A. (2016). The impact
of incongruous lake temperatures on regional climate extremes downscaled from the
CMIP5 archive using the WRF model. *Journal of Climate, 29*(2), 839-853.
Stepanenko, V., & Lykossov, V. (2005). Numerical modeling of heat and moisture transfer
processes in a system lake-soil. *Russ. Meteorol. Hydrol, 3*, 95-104.
Stepanenko, V., Machul'Skaya, E., Glagolev, M., & Lykossov, V. (2011). Numerical modeling
of methane emissions from lakes in the permafrost zone. *Izvestiya, Atmospheric and*
*Oceanic Physics, 47*, 252-264.
Stepanenko, V. M., Goyette, S., Martynov, A., Perroud, M., Fang, X., & Mironov, D. (2010).
First steps of a lake model intercomparison project: LakeMIP. *Boreal environment*
*research, 15*(2), 191.
Subin, Z. M., Riley, W. J., & Mironov, D. (2012). An improved lake model for climate
simulations: Model structure, evaluation, and sensitivity analyses in CESM1. *Journal*
*of Advances in Modeling Earth Systems, 4*(1).
doi:https://doi.org/10.1029/2011MS000072
Sun, L., Liang, X.-Z., & Xia, M. (2020). Developing the Coupled CWRF-FVCOM Modeling
System to Understand and Predict Atmosphere-Watershed Interactions Over the Great
Lakes Region. *Journal of Advances in Modeling Earth Systems, 12*(12),
e2020MS002319. doi:https://doi.org/10.1029/2020MS002319
Todorovich, P. (2009). America's emerging megaregions and implications for a national growth
strategy. *International Journal of Public Sector Management, 22*(3), 221-234.
Vaccaro, L., & Read, J. (2011). *Vital to Our Nation's Economy: Great Lakes Jobs*. Retrieved
from https://www.michiganseagrant.org/wp-content/uploads/2018/10/11-203-Great-
Lakes-Jobs-report.pdf
Valcke, S., Redler, R., Budich, R., Valcke, S., & Redler, R. (2012). The oasis coupler. *Earth*
*System Modelling-Volume 3: Coupling Software and Strategies*, 23-32.
Wang, J., Bai, X., Hu, H., Clites, A., Colton, M., & Lofgren, B. (2012). Temporal and Spatial
Variability of Great Lakes Ice Cover, 1973–2010*. *Journal of Climate, 25*(4), 1318-





1329. doi:https://doi.org/10.1175/2011jcli4066.1

Wang, J., Xue, P., Pringle, W., Yang, Z., & Qian, Y. (2022). Impacts of Lake Surface
Temperature on the Summer Climate Over the Great Lakes Region. *Journal of*
*Geophysical Research: Atmospheres, 127*(11), e2021JD036231.
doi:https://doi.org/10.1029/2021JD036231

Woolway, R. I., Anderson, E. J., & Albergel, C. (2021). Rapidly expanding lake heatwaves
under climate change. *Environmental Research Letters, 16*(9), 094013.

Xiao, C., Lofgren, B. M., Wang, J., & Chu, P. Y. (2016). Improving the lake scheme within a
coupled WRF-lake model in the Laurentian Great Lakes. *Journal of Advances in*
*Modeling Earth Systems, 8*(4), 1969-1985. doi:https://doi.org/10.1002/2016MS000717

Xue, P., Pal, J. S., Ye, X., Lenters, J. D., Huang, C., & Chu, P. Y. (2017). Improving the
Simulation of Large Lakes in Regional Climate Modeling: Two-Way Lake–
Atmosphere Coupling with a 3D Hydrodynamic Model of the Great Lakes. *Journal of*
*Climate, 30*(5), 1605-1627. doi:https://doi.org/10.1175/jcli-d-16-0225.1

Xue, P., Schwab, D. J., & Hu, S. (2015). An investigation of the thermal response to
meteorological forcing in a hydrodynamic model of Lake Superior. *Journal of*
*Geophysical Research: Oceans, 120*(7), 5233-5253.
doi:https://doi.org/10.1002/2015JC010740

Xue, P., Ye, X., Pal, J. S., Chu, P. Y., Kayastha, M. B., & Huang, C. (2022). Climate
projections over the Great Lakes Region: using two-way coupling of a regional climate
model with a 3-D lake model. *Geosci. Model Dev., 15*(11), 4425-4446.
doi:10.5194/gmd-15-4425-2022

Ye, X., Chu, P. Y., Anderson, E. J., Huang, C., Lang, G. A., & Xue, P. (2020). Improved
thermal structure simulation and optimized sampling strategy for Lake Erie using a data
assimilative model. *Journal of Great Lakes Research, 46*(1), 144-158.
doi:https://doi.org/10.1016/j.jglr.2019.10.018

Yeates, P., & Imberger, J. (2003). Pseudo two-dimensional simulations of internal and
boundary fluxes in stratified lakes and reservoirs. *International Journal of River Basin*
*Management, 1*(4), 297-319.

Zhong, Y., Notaro, M., Vavrus, S. J., & Foster, M. J. (2016). Recent accelerated warming of the
Laurentian Great Lakes: Physical drivers. *Limnology and Oceanography, 61*(5), 1762-
1786. doi:https://doi.org/10.1002/lno.10331
