# Peer review of "Enhancing Winter Climate Simulations of the Great Lakes: Insights from a New Coupled Lake-Ice-Atmosphere (CLIAv1) Model on the Importance of Integrating 3D Hydrodynamics with a Regional Climate Model"

_Geoscientific Model Development, 2024_

## Referee Comment (RC2)

**Review Comments**

Xue et al. developed a coupled lake-ice-atmosphere modeling system of NU-WRF/FVCOM. The new model demonstrates clear advantages over the 1D lake model (LISSS). The authors also address a crucial question regarding the key processes influencing lake thermal structure and ice cover in 3D lake models through well-designed numerical experiments. The overall work is strong, and the process analysis is comprehensive. The manuscript requires adjustments to its structure and presentation for clarity and consistency. Below are specific comments and suggestions for improvement:

1. On Line 30, add the full name of "LSTs" (presumably "lake surface temperatures") upon first mention for clarity.

2. In Figure 1, the blue line for "FVCOM mesh" does not appear to be visible in panel (a). Consider using blue in panel (b) instead of red to clearly show the FVCOM mesh. Additionally, add the names of the lakes to panel (b) for better context.

3. For all figures, it is standard practice to label subplots with (a), (b), (c), etc. Please add these labels to improve readability.

4. In Section 3.2, clarify whether the NU-WRF/LISSS configuration uses the same lake mesh as shown in Figure 1b (like NU-WRF/FVCOM). This will help readers understand the setup differences between the two models.

5. In Figure 2, observations from GLSEA show some spikes in temperature and ice cover time series (e.g., Lake Ontario's low-temperature spike in February and ice cover spike in February), but the simulations appear smoother. Could the authors explain this discrepancy? Is it due to model limitations or data processing?

6. In Figure 3, while Lake3D performs much better than Lake1D, the spatial pattern in GLSEA observational data is still more heterogeneous compared to the Lake3D simulation. What are the potential reasons for this? Additionally, were any parameters tuned, or initial conditions adjusted to improve the Lake3D simulation compared to Lake1D? If so, please clarify.

7. In Figure 6, what are the potential reasons for the underestimation of latent heat flux by Lake3D over Spectacle Reef? Please discuss possible causes.

8. The C2-related analysis is currently included in the discussion section (Section 5), which is unusual. This content should be moved to the results section. The discussion section should focus on synthesizing findings from both C1 and C2 experiments rather than presenting new results. The C2 experiments are important and should not be overlooked or buried in the discussion.

9. The explanation of equations in Section 5.2/5.3 would be better placed in the methods section, maybe in the experiment design subsection for C2 experiments. This would improve the flow and readability of the manuscript.

10. For the C2 experiments, it would be valuable to include analysis of sensible/latent heat, T2, and wind speed comparisons for the different physics turnoff experiments. This would provide a more comprehensive understanding of the impacts on lake-atmosphere interactions. If space is limited, consider adding this analysis as supplementary material.

---

## Author Comment (AC1)

Reviewer 1:

General comments

This paper describes model improvements when using a fully 3-dimensional hydrodynamic model within a regional climate model. The authors describe the simulation improvements with the 3D model as compared with a 1D model, and explore the physical processes that lead to this improvement. As the authors acknowledge, this coupling has been performed before and several other papers have highlighted the importance of improving deep lake representation. The novelty in this paper is that they explore the physical reasons as to why these improvements occur. The results in the latter half of the paper are interesting, and the authors do a thorough job explaining the physical processes underlying the model improvement.

My main comments are surrounding the paper organization, figure clarity, and being sure to accurately acknowledge prior work in this space. These revisions are relatively minor, and I recommend publication with this minor changes.

Thank you for your thoughtful comments and suggestions to help improve our manuscript. We have carefully considered your feedback and incorporated the suggested revisions into the updated manuscript, along with a detailed point-by-point response to facilitate your evaluation. For your convenience, we have also included a version of the manuscript with tracked changes.

Process description

> The process-level description that the authors are highlighting isn't explained with equations until much later in the paper (5.2 for the heat transport, 5.3 for the vertical mixing). It would have helped if the authors had used a more traditional framework and described the important component models upfront (e.g., in Section 2.2) to make all the processes clear before getting to the results
> In the same way, there is one paragraph on the 1D model in Section 2.2 that is out of place. Given that the paper focuses on how the results are so different, more time to clarify the key difference of the model (in equations) would have set up the paper better.
> This would be helpful for later interpretation, e.g., Section 5.1 – line 523-4 states "Instead, only ice thermal dynamics are simulated as in the 1D lake model."
> Something that describes this 1D process would be helpful for the reader.

Response: As suggested, we have revised the manuscript structure to follow a more traditional framework and now describe the key component models up front in Section 2.2. We have also added a new Section 2.3 to describe the 1D lake model (including equations) to provide readers with the necessary background information.

Figure consolidation – Many of the figures have redundant information in them, e.g.,

> Figures 4/10/13 (vertical T profiles at the Spectacle Reef Site): Many duplicate panels in these three figures. While I understand the intention to step through the different experiments, I often wanted to see these figures side by side. I think these panels could be effectively combined to make one comprehensive figure.

> Figure 5/8/14 (spatial distribution of ice cover): Same as above – lots of redundant information on these figures, and it would help the reader to see some consolidation here.

**Response:** Thank you for your suggestion regarding Figures 4, 10, and 13 (vertical temperature profiles at the Spectacle Reef site) and Figures 5, 8, and 14 (spatial distribution of ice cover). We understand your preference for a consolidated view to facilitate side-by-side comparison across experiments.

To address this helpful suggestion, we have compiled the relevant figures into **Supplementary Figures S4 and S5**, enabling direct visual comparison of results across the different experimental setups.

We have chosen to retain **original Figure 4** and have combined **Figures 10 and 13 into a new Figure 10, as you suggested,** in the main manuscript, as this structure provides a step-by-step narrative aligned with the logical progression of our analysis. This sequential approach helps guide the reader through the key hydrodynamic processes and their individual impacts at each stage of the study.

Since **Figures 5, 8, and 14** each represent different phases of the experimental analysis, we feel it is not ideal to present them together early in the manuscript without discussion until much later. Maintaining their original structure supports a clearer, more coherent progression for the reader. We also simplified **Figure 14** (now **Figure 13**) to remove redundant information.

That said, for readers interested in **side-by-side comparisons** of thermal structure and ice cover across model cases, we provide a consolidated view in **Supplementary S4 and S5**, as you suggested.

We believe this approach—maintaining a logically structured main figure layout while offering a complementary, reformatted comparison in the supplementary materials—offer an optimal balance between clarity, readability, and comparative utility.

> All figure panels could use some additional labeling on rows/columns, as they change from figure to figure (e.g., sometimes the different months are the rows, sometimes

they are the columns). Also, many of the fonts and legends are *extremely* small and hard to read (e.g. Figures 9, 11, 12).  Some sublabels (e.g., labeling the panels a, b, c, etc.) would help to connect specific figures to the text.

**Response:** Regarding figure quality, labeling, and readability, we have carefully reviewed and revised all figures based on your suggestions. These adjustments improve clarity and visual coherence across the manuscript, ensuring a more accessible and effective presentation of the data. We appreciate your suggestions, which helped us improve the clarity and accessibility of our visual materials.

References to prior work: The authors do acknowledge that some work has been done in this space before, but I don't think that they have fully acknowledged all that has been done in the regional climate community, e.g.a few key ones that are missing include

1. Leon et al. 2007 ELCOM in the Canadian regional model (CRCM)
2. Turuncoglu et al. 2013 ROMS in the Regional climate model (RegCM)
3. Bryan et al. 2015, showing the impacts of 1D lakes in RegCM

**Response:** Thank you for bringing to our attention the omission of several key studies. We have now incorporated additional references in the Introduction to reflect important prior work by the regional climate modeling community.

**Leon, L. F., Lam, D., Schertzer, W., and Swayne, D.** (2005): Lake and climate models linkage: a 3-D hydrodynamic contribution, Adv. Geosci., 4, 57–62, https://doi.org/10.5194/adgeo-4-57-2005, 2005.

**Leon, L. F., Lam, D. C. L., Schertzer, W. M., Swayne, D. A., & Imberger, J.** (2007). Towards coupling a 3D hydrodynamic lake model with the Canadian regional climate model: simulation on Great Slave Lake. Environmental Modelling & Software, 22(6), 787-796.

**Turuncoglu, U. U., Giuliani, G., Elguindi, N., and Giorgi, F.**(2013): Modelling the Caspian Sea and its catchment area using a coupled regional atmosphere-ocean model (RegCM4-ROMS): model design and preliminary results, Geosci. Model Dev., 6, 283–299, https://doi.org/10.5194/gmd-6-283-2013, 2013.

**Bryan A. M., A. L. Steiner, and D. J. Posselt** (2015), Regional modeling of surface-atmosphere interactions and their impact on Great Lakes hydroclimate, *J. Geophys. Res. Atmos.*, 120, 1044–1064, doi:10.1002/2014JD022316.

Minor editorial:

1. Line 35-36:  change to "…. this study identified the key processes influencing…."

**Response:** Changed.

2. Lines 153-160: This is a long list of possible obstacles followed by a long list of references. Could the authors parse this out more so there is more clarity in which studies were investigating specific components?

**Response:** Good suggestion, revised as "*The lack of fully resolved lake hydrodynamics in models (Xue et al., 2017; Sharma et al., 2018), including lake circulation (Song et al., 2004), upwelling and downwelling, thermal bar formation (Martynov et al., 2010, 2012), explicit horizontal mixing, and ice motion, along with overly simplified stratification processes (Bennington et al., 2014), unrealistic treatment of eddy diffusivity (Stepanenko et al., 2010; Gu et al., 2015; Mallard et al., 2015)  has been the main obstacle in further improving climate simulations for the Great Lakes Basin.*"

3. Line 179: LISSS, first use of acronym with no description.

**Response:** Added.

4. Suggest to remove Table 1 as this information is all in the text

**Response:** We believe that Table 1 plays a crucial role in providing a clear and concise overview of our experimental design, particularly given the large number of sensitivity experiments conducted. The table enables readers to quickly understand the structure, variations, and purpose of each experiment without having to extract this information from the main text. Given its importance in improving the clarity and accessibility of our methodology, we respectfully request to retain Table 1 in the manuscript.

5. Figure 2 – why not include the 1D model on here as well for comparison?

**Response:**  We see your point—this information has now been added to Supplementary Figure S1. In Figure 2, we focus specifically on evaluating the performance of the 3D lake model against observations.

6. Line 460 – can you show these observation locations on one of the spatial figures? Also, you note they are selected because of the highest ice coverage – what about observations on Lake Erie?  That usually tends to be the most ice covered.

**Response:**  We have included the observation locations in Figure 1b.

[Figure]

**Fig 1b in the manuscript:** The two dots denote the locations of Granite Island (87.4°W, 46.7°N) on Lake Superior and Spectacle Reef (84.1°W, 45.7°N) on Lake Huron. The triangle marker denotes the location (82.58°W,45.16°N) of thermistor observation in deep, central Lake Huron, where the water depth is 220 meters.

We appreciate the opportunity to clarify our methodology and selection criteria. The original text was imprecise, and we have revised it to read:
 **"Lake Superior and Lake Huron were selected for demonstration because studies have shown that deeper, larger Great Lakes present more complex hydrodynamic challenges that 1D models consistently fail to represent accurately, often resulting in substantial errors."**

While Lake Erie does experience significant ice coverage, it is relatively shallow (with an average depth of approximately 19 meters). The performance of 1D lake models in Lake Erie has been mixed, with reasonable accuracy achievable in some cases through empirical tuning and careful configuration. This has been demonstrated in several studies (Martynov et al., 2010; Subin et al., 2012; Bennington et al., 2014; Gu et al., 2015). In contrast, numerous studies have shown that deeper, larger Great Lakes—such as Lake Superior and Lake Huron—pose greater hydrodynamic complexity, which 1D models consistently struggle to capture, often leading to substantial simulation errors.

Therefore, we focused our comparative analysis on Lake Superior and Lake Huron, where the limitations of 1D lake models are more clearly expressed and consistently observed. We believe

the revision helps to clarify our rationale and better align our site selection with the study's objectives.

7. Figure 7 – These line plots are very hard to read. Could it have the observations on one y axis, and the two model versions bias on a second y axis?

**Response**: We appreciate your suggestion and have generated the revised figure as recommended. After comparison, we find that both the original and revised versions have their respective strengths and limitations. The original figure is more effective at illustrating how the model tracks observed variability—particularly for air temperature—though it is somewhat harder to interpret for wind speed. In contrast, the revised figure presents model biases relative to observations more clearly, but it lacks a direct visualization of the model's ability to capture temporal variability, as it only shows the biases.

To address this, we have decided to include both versions. The original Figure 7 has been refined with improved resolution, and the revised bias-focused plot has been included as Supplementary Figure S3.

Both figures are provided here for your convenience.

[Figure]

**Figure 7 in the manuscript.** Time series of daily air temperature (°C, upper panels; a, b) at 2-m height (T2) and wind speed (m/s, lower panels; c, d) from GLEN observations (black lines), NU-WRF/FVCOM 3D lake model simulations (red lines), and NU-WRF/LISSSS 1D lake model simulations (blue lines) at Granite Island on Lake

Superior and Spectacle Reef on Lake Huron during November 2014-March 2015. The RMSE and temporal correlations between the simulations and GLEN observations are provided in each panel.

[Figure]

**Figure S3.** Time series of daily 2-meter air temperature (°C; upper panels: a, b) and wind speed (m/s; lower panels: c, d) from GLEN observations (black lines, left y-axis) at Granite Island (Lake Superior) and Spectacle Reef (Lake Huron) during November 2014 to March 2015. Model biases relative to observations are shown on the right y-axis for the NU-WRF/FVCOM simulation with the 3D lake model (red lines) and the NU-WRF/LISSS simulation with the 1D lake model (blue lines).

8. Line 574 – change "… GLSEA is not able to well capture…." to "…. GLSE cannot capture…"

**Response:** The sentence has bee removed.

9. Figure 11 – this figure is so hard to read, yet seems to be a very important one. Note that dT/dt is the black line in the caption. It is very hard to see the difference between the black and the purple line, so perhaps dash one of them.

**Response:** Thank you for the suggestion. We have revised the figure accordingly, and it is now much clearer. The updated figure is provided here for your convenience.

[Figure]

**Figure 11 in the manuscript.** `Monthly-averaged vertical profiles of key terms in the temperature equation in the Cl-l (Lake3D) simulation at the deep-water thermistor site (220 m) in central Lake Huron (site location is on Fig. 1) from November 2014 through March 2015. The black line represents the temperature change rate ($\partial T/\partial t$), while the dashed blue and magenta lines represent the contributions from 3D advection and vertical diffusion, respectively. Horizontal diffusion is omitted here due to its negligible contribution throughout the winter season.`

10. Lines 620-623: Can you comment more on the importance and implications of this issue?

**Response:** The entire paragraph has been rewritten to more clearly elaborate the role of advective heat transport, with coherent reference to Figures 10 and 11.

11. Line 632: what specifically is meant by "sophisticated"? At this point, most readers would want to know specifically which model is used (with a reference).

**Response:** As you previously suggested, we have moved the model description to Section 2.2. Specifically, we now clarify that the simulation utilizes the **Mellor–Yamada Level 2.5 turbulence closure scheme**, a widely used approach for geophysical fluid dynamics (Mellor and Yamada, 1982).

**Reference:** Mellor, G. L., & Yamada, T. (1982). Development of a turbulence closure model for geophysical fluid problems. *Reviews of Geophysics*, 20(4), 851–875.

12. Figure 12 – there are six terms in equation 3, yet only five are shown in the plot. Is there a reason why the horizonal diffusion is not plotted? If too small, then this should be included in the caption or the text.

**Response:** Thank you for your observation. The horizontal diffusion term, along with a few other terms, is indeed also small, making them difficult to visualize separately in the plot. To improve clarity, we have revised the figure (included below for your convenience) and represented these terms using a dashed line for better visibility and have explicitly noted its small magnitude in the figure caption to ensure transparency.

We note that the primary purpose of this figure is to highlight the dominant terms in the temperature tendency and turbulence kinetic energy (TKE) budget. In particular, we emphasize that shear production, a key source term for TKE, is largely balanced by eddy dissipation, the primary sink term. This balance underscores the importance of resolving realistic flow conditions, which directly govern shear-driven mixing.

[Figure]

**Figure 12. in the manuscript.** Monthly averaged vertical profile of each term of the turbulence kinetic equation in the C1-1 (Lake3D) simulation at the deep-water thermistor site (220 m) in central Lake Huron (site location is on Fig. 1) from November 2014 through March 2015. The profiles include shear production (green), buoyancy production (cyan dashed), vertical diffusion of TKE (magenta), dissipation rate (black), 3D advection of TKE (red dashed), and the TKE change rate term $\partial q^2/\partial t$ (blue dotted). Shear production is the dominant source term balancing the dissipation rate (sink), while the other terms—buoyancy production, 3D advection—are comparatively smaller in magnitude. The dominance of shear-driven mixing emphasizes the importance of resolving realistic current structures.

Thank you for your time and effort in helping us improve the manuscript. We hope that our responses and revisions have satisfactorily addressed your concerns.

---

## Author Comment (AC4)

Reviewer 2:

**Review Comments**

Xue et al. developed a coupled lake-ice-atmosphere modeling system of NU-WRF/FVCOM. The new model demonstrates clear advantages over the 1D lake model (LISSS). The authors also address a crucial question regarding the key processes influencing lake thermal structure and ice cover in 3D lake models through well-designed numerical experiments. The overall work is strong, and the process analysis is comprehensive. The manuscript requires adjustments to its structure and presentation for clarity and consistency. Below are specific comments and suggestions for improvement:

Response: Thank you for your constructive comments and suggestions to improve our manuscript. We have carefully revised the manuscript, taking all of your feedback into full consideration. A point-by-point response is provided below to facilitate your evaluation. For your convenience, we have also included a tracked-changes version of the revised manuscript.

*We recognized the need to communicate and emphasize the purpose of the study and the research questions more clearly to a broader audience. To facilitate your evaluation, we appreciate the opportunity to elaborate on the subject further here for your reference.*

**Background:** There has been *a well-established consensus* in the Great Lakes regional modeling community that climate models used in this region should incorporate lake simulations—particularly three-dimensional (3D) lake models when available (Briley & Jorns, 2021). Numerous studies, dating back to the early 2010s have acknowledged the limitations of such models when applied to large, deep lakes like those in the Great Lakes system. These limitations, primarily due to the simplified representation of lake hydrodynamics, have led many researchers (e.g., Martynov et al., 2010, 2012; Stepanenko et al., 2010; Bennington et al., 2014; Gu et al., 2015; Mallard et al., 2015) to advocate for the use of 3D lake models in future coupled RCM applications.

In alignment with this, more recent studies employing RCMs coupled with 3D lake models have consistently shown improved performance and predictive skill in the Great Lakes. These advancements have been documented in the peer-reviewed literature (Xue et al., 2017; Sun et al., 2020) and the Great Lakes Climate Modeling Workshop reports (Briley & Jorns, 2021).

Therefore, it is neither our intention to reiterate an already well-established consensus. This study is not driven by an effort to improve 1D lake models or to refine 3D lake models further. **Rather, our focus is on understanding the fundamental hydrodynamic processes absent in 1D models and how they are resolved in 3D frameworks.**

We clarify this in the revised text in the new Discussion Section line 800-807: "*Within this context, the core contribution of this research lies in advancing our understanding of a central*

*question in Great Lakes regional climate modeling: What are the key **hydrodynamic processes missing** from one-dimensional (1D) lake models—processes that are critical for simulating lake thermal structure and ice cover during the cold season—and how are these processes resolved in three-dimensional (3D) lake models? Our findings provide **generalized insights that are not dependent** on specific model configurations, tuning strategies, or the reproduction of individual observed events, making them broadly applicable across different modeling systems and lake conditions.*"

To answer this, we structured the study in two stages:

- **C-1: Foundational Evaluation of the Coupled System** In this component, we conduct benchmark simulations using NUWRF coupled with FVCOM (3D Lake) and NUWRF coupled with a LISSS (1D lake model). The results aim to verify the skill of our coupled modeling NUWRF-FVCOM system in reproducing observed lake surface temperatures (LST) and ice cover. This foundational experiment serves *not* to rehash the well-known limitations of 1D models, but to **establish confidence in the coupled NUWRF-FVCOM framework** and justify its application for process-level investigation in the next stage (C-2 experiments).

- **C-2: Diagnosing the Hydrodynamic Processes Missing in 1D Lake Models** This represents the core contribution of our study and distinguishes it from previous work, including our own earlier efforts using coupled RCM–3D lake models. We systematically identify three key hydrodynamic processes that are absent in 1D lake models but are resolved in 3D models, which account for the improved cold-season performance in simulating LST and ice cover: 1) Lateral ice transport; 2) Advective heat transport; 3) Shear-induced turbulence. Critically, all three of these processes are dynamically linked to water currents—spatially and temporally evolving flow fields that are fundamentally unresolved or crudely simplified in 1D lake models. This represents the key scientific insight of our study: these dominant hydrodynamic processes responsible for realistic wintertime lake thermal structure and ice cover are current-driven, and thus structurally absent in 1D models. This finding constitutes the key contribution of our work and, we believe, offers an important step forward in understanding why 3D models perform better—not merely that they do.

So we structured and revised our manuscript centered on the following key criteria: (1) Does the study pose a meaningful central scientific question that benefits the broader research community? (2) Does the manuscript present robust evidence in support of its conclusions? (3) Are the conclusions well-supported and persuasive to readers?

Thank you again for your time and effort in providing comments and suggestions and for evaluating our revision. Below is our point-by-point response, guided by the revision principles and criteria we outlined and your suggestions.

1. On Line 30, add the full name of "LSTs" (presumably "lake surface temperatures") upon first mention for clarity.

**Response:** Thanks, the missing full name is added.

2. In Figure 1, the blue line for "FVCOM mesh" does not appear to be visible in panel (a). Consider using blue in panel (b) instead of red to clearly show the FVCOM mesh. Additionally, add the names of the lakes to panel (b) for better context.

**Response:** Figure 1 has been revised as you suggested to enhance clarity and improve visibility. We include it here for your convenience.

[Figure]

**Figure 1 in the manuscript.** NU-WRF nested domains (upper panel) and unstructured mesh used in FVCOM to represent the Great Lakes in FVCOM (lower panel). The two dots denote the locations of Granite Island (87.4°W, 46.7°N) on Lake Superior and Spectacle Reef (84.1°W, 45.7°N) on Lake Huron. The triangle marker denotes the location (82.58°W,45.16°N) of thermistor observation in deep, central Lake Huron, where the water depth is 220 meters.

3. For all figures, it is standard practice to label subplots with (a), (b), (c), etc. Please add these labels to improve readability.
**Response:** We have now added labels to all subplots across the figures, as recommended.

4. In Section 3.2, clarify whether the NU-WRF/LISSS configuration uses the same lake mesh as shown in Figure 1b (like NU-WRF/FVCOM). This will help readers understand the setup differences between the two models.

**Response:** No, LISSS and FVCOM do not use the same mesh, as their integration with the atmospheric model (NUWRF) differs fundamentally—both in terms of architecture and representation of lake dynamics.In short, LISSS has to operate on the NUWRF atmospheric model grid, whereas FVCOM uses its own unstructured grid that is independent of the NUWRF grid.

This has now been explicitly explained and clarified in section 3 in revision as "*FVCOM is a complex, fully prognostic 3D hydrodynamic model. It operates on its own unstructured mesh, which is independent of the NUWRF atmospheric grid, and is well-suited to resolving complex lake geometry, shorelines, and bathymetry. Therefore, coupling between NUWRF and FVCOM must be achieved through an external coupler, which facilitates end-to-end, two-way exchange of information at any desired interval. NU-WRF and FVCOM are run simultaneously, exchanging information bidirectionally at 1-hour intervals through the OASIS3-MCT coupler. FVCOM dynamically calculates the LST and ice cover, providing these as overlake surface boundary conditions to NU-WRF. Meanwhile, NU-WRF calculates and supplies the atmospheric forcings required by FVCOM, including surface air temperature, surface air pressure, relative and specific humidity, total cloud cover, surface winds, and downward shortwave and longwave radiation.  No tuning was applied to FVCOM in the coupled configuration to improve consistency with observations, as the default FVCOM configuration was applied.*

*In contrast, 1D lake models, including LISSS, are simplified, column-based lake models directly embedded within NUWRF without using a coupler. Each NUWRF atmospheric grid cell over a lake surface contains one corresponding vertical water column 1D model, which simulates thermal processes in the vertical direction. Collectively, these columns provide a pseudo-3D representation of the lake but do not simulate horizontal processes such as advection, circulation, or lateral ice transport. As a result, LISSS must use the same horizontal resolution as the NUWRF grid.*"

5. In Figure 2, observations from GLSEA show some spikes in temperature and ice cover time series (e.g., Lake Ontario's low-temperature spike in February and ice cover spike in February), but the simulations appear smoother. Could the authors explain this discrepancy? Is it due to model limitations or data processing?

**Response:** Thank you for your comments. We address Comments 5 and 6 together in the response below.

6.     In Figure 3, while Lake3D performs much better than Lake1D, the spatial pattern in GLSEA observational data is still more heterogeneous compared to the Lake3D simulation. What are the potential reasons for this? Additionally, were any parameters tuned, or initial conditions adjusted to improve the Lake3D simulation compared to Lake1D? If so, please clarify.

**Response:** We appreciate the reviewer's observation. Indeed, Figure 3 shows that while the Lake3D simulation captures the large-scale thermal structure of the Great Lakes with good fidelity—particularly when compared to the 1D model—there is a noticeable difference in the degree of spatial heterogeneity when compared to the GLSEA (Great Lakes Surface Environmental Analysis) observational product. We have now explicitly acknowledged these model limitations in the revised text in line 454-459 as "*While the model successfully reproduces the overall seasonal evolution, it misses some episodic fluctuations. For example, observational data from GLSEA show short-term spikes in both temperature and ice cover—such as the notable low-temperature and ice cover spikes in Lake Ontario during February—that are not fully captured in the simulation. The modeled LST and ice cover time series tend to appear smoother than the observations (Fig 2. a4, b4) ..... Also, the spatial pattern in GLSEA observational data appears more heterogeneous on a finer scale compared to the 3D Lake simulation.*"

There are several factors contributing to this discrepancy:

**Resolution Differences:** The GLSEA product is generated from satellite-derived data with a horizontal resolution of approximately 1.3 km, providing fine-scale spatial detail in open-water areas. In contrast, while FVCOM's unstructured grid allows for flexible mesh refinement, the configuration used in this study employed a coarser horizontal resolution of up to 4 km in open lake regions. This coarser resolution may smooth out finer-scale spatial variability, reducing the ability of the model to replicate small-scale thermal features present in the observations.

**Model Limitations:** While the 3D lake model substantially improves the representation of physical processes compared to the 1D model, it is not without limitations. Like all process-based models, both NUWRF and FVCOM are subject to uncertainties in boundary conditions and internal dynamics, subgrid parameterizations, which can limit its ability to fully capture observed spatial variability. These limitations are also relevant to the issue raised in Comment #5, where we believe that the coupled model may not have adequately resolved the episodic events responsible for the observed temperature spikes.

That said, these small-scale spatial mismatches are not central to the main objective of this study, *as elaborated at the beginning of our response.* Our primary focus is to **identify the key hydrodynamic mechanisms absent from 1D lake models that contribute to their cold-season temperature biases and excessive ice cover**. The Lake3D simulation's demonstrated ability to reproduce large-scale spatial and temporal patterns provides a robust foundation for the mechanistic analysis and conclusions presented in this work.

We have also explicitly clarified in lines 361-363 as "*No tuning was applied to FVCOM in the coupled configuration to improve consistency with observations, as the default FVCOM configuration was applied.*" The FVCOM component used in this study employed a standard parameter configuration that has been widely applied and documented in prior publications. This configuration has been fully archived and made publicly accessible via Zenodo to ensure transparency and reproducibility. As we noted in the original manuscript, the initial lake conditions of November 2014 were obtained from FVCOM standalone simulations driven by Climate Forecast System Reanalysis (CFSR) forcing.

For the 1D lake model, we used LISSS in its optimal configuration, as determined by Notaro et al. (2023), who conducted a comprehensive sensitivity analysis across more than 20 configurations. We adopted the best-performing setup reported in their study to ensure a fair and representative comparison between the 1D and 3D models. However, as explained in the beginning, we acknowledge that 1D lake models can be tuned to better match observational data (e.g., through lumped eddy diffusivity; Xiao et al., 2016; Bennington et al., 2014), yet this is directly related to our central research question.

*Importantly, our central conclusion is that the performance differences arise from the **presence or absence of key physical processes** as generalized insights that are not dependent on specific model configurations, tuning strategies, or the reproduction of individual observed events, making them broadly applicable across different modeling systems and lake conditions*

We hope this clarifies both our methodology and the core scientific contribution of the study.

7. In Figure 6, what are the potential reasons for the underestimation of latent heat flux by Lake3D over Spectacle Reef? Please discuss possible causes.

**Response**: Good question. We have specifically discussed this in the revised manuscript in lines 557-569 with supplementary Fig S2 as "*Latent heat in Spectacle Reef is the only exception, where NU-WRF/FVCOM struggles to capture the magnitude of the upward latent heat flux due to the overestimated ice cover at the site (Fig. S2). Ice cover plays a critical role in modulating latent heat exchange: in the bulk aerodynamic formulation, latent heat flux is scaled by the open water fraction, as ice acts as a physical barrier to evaporation and moisture transfer. A higher modeled ice fraction reduces the effective evaporation area, resulting in suppressed moisture exchange and, consequently, underestimation of latent heat flux. As shown in Fig. S2, the model substantially overestimates ice cover at this site in January and maintains high ice concentration through February. This persistent overestimation directly reduces the open water fraction, contributing to low latent heat. Interestingly, the observed latent heat flux remains elevated in February despite observed ice cover approaching 90%. This apparent discrepancy suggests potential uncertainty in either the observed ice cover, the latent heat flux measurements, or both, and warrants further investigation.*"

[Figure]

**Figure S2.** Time series of ice cover at Spectacle Reef for the 2014–2015 winter season, comparing observations from the Great Lakes Surface Environmental Analysis (GLSEA, blue, which has ice data sourced from the National Ice Center) with the Lake3D model simulation (red).

8. The C2-related analysis is currently included in the discussion section (Section 5), which is unusual. This content should be moved to the results section. The discussion section should focus on synthesizing findings from both C1 and C2 experiments rather than presenting new results. The C2 experiments are important and should not be overlooked or buried in the discussion.
**Response:** Agreed. we have now relocated and retitled this section as result section 4.2 "Diagnosing the Key Hydrodynamic Processes Missing in 1D Lake Models"

9. The explanation of equations in Section 5.2/5.3 would be better placed in the methods section, maybe in the experiment design subsection for C2 experiments. This would improve the flow and readability of the manuscript.

**Response:** As suggested, we have revised the manuscript structure to follow a more traditional framework and now describe the key component models up front in Section 2.2.

10. For the C2 experiments, it would be valuable to include analysis of sensible/latent heat, T2, and wind speed comparisons for the different physics turnoff experiments. This would provide a more comprehensive understanding of the impacts on lake-atmosphere interactions. If space is limited, consider adding this analysis as supplementary material.

**Response:** We sincerely thank you for this thoughtful comment. We both agree that looking at sensible and latent heat fluxes along with near-surface atmospheric variables (like air temperature at 2 meters and wind speed) can help us learn more about how lake processes affect the atmosphere above them. However, we respectfully clarify that the primary goal of the **C2 experiment** is more narrowly focused in both scope and contribution, as elaborated at the beginning of our response. We note in lines 608-611 "*Note that in the discussion of the C2 experiments below, analyses are focused on the major 3D lake processes that influence the simulated limnological patterns of lake temperature and ice cover, not the overlying atmospheric conditions, which are beyond the scope of this study.*"

We also acknowledge that, due to project closeout and the expiration of our HPC allocations, we were unable to archive the complete atmospheric datasets. As this study focuses specifically on hydrodynamic processes within the lake system, we made a strategic decision to prioritize diagnostics directly relevant to lake thermal structure and ice cover. Re-running the full coupled system to extract atmospheric fields would require significant computational resources and is currently beyond our capacity. However, we believe the process-level analysis and evidence we have presented are robust and sufficiently address the central scientific question.

Thank you for your time and effort in helping us improve the manuscript. We hope that our responses and revisions have satisfactorily addressed your concerns.